# Electrospray-assisted cryo-EM sample preparation to mitigate interfacial effects

Zi Yang [1,2,4], Jingjin Fan[3,4], Jia Wang [1,2], Xiao Fan[1,2], Zheng Ouyang [3] ✉, Hong-Wei Wang [1,2] ✉ & Xiaoyu Zhou [3] ✉

Addressing interfacial effects during specimen preparation in cryogenic electron microscopy remains challenging. Here we introduce ESI-cryoPrep, a specimen preparation method based on electrospray ionization in native mass spectrometry, designed to alleviate issues associated with protein denaturation or preferred orientation induced by macromolecule adsorption at interfaces. Through fine-tuning spraying parameters, we optimized protein integrity preservation and achieved the desired ice thickness for analyzing target macromolecules. With ESI-cryoPrep, we prepared high-quality cryo-specimens of five proteins and obtained three-dimensional reconstructions at near-atomic resolution. Our findings demonstrate that ESI-cryoPrep effectively confines macromolecules within the middle of the thin layer of amorphous ice, facilitating the preparation of blotting-free vitreous samples. The protective mechanism, characterized by the uneven distribution of charged biomolecules of varying sizes within charged droplets, prevents the adsorption of target biomolecules at air–water or graphene–water interfaces, thereby avoiding structural damage to the protein particles or the introduction of dominant orientation issues.

Cryogenic electron microscopy (cryo-EM), a transmission electron microscopy technique that examines cryogenically preserved hydrated specimens, has emerged as a powerful tool for analyzing macromolecular structures in recent years. Recent advancements in both instrumentation and software, including the use of direct electron detection cameras[1] and the application of maximum a posteriori processing algorithms[2,3] for three-dimensional (3D) reconstruction, have led to a 'resolution revolution'[4] in cryo-EM, enabling the direct dissection of atomic-level biological mechanisms through high-resolution 3D density maps. One notable advantage of cryo-EM is its ability to determine macromolecular structures in their more close-to-native states and requires a smaller number of samples compared to other structural biology methods[5]. This is primarily attributed to the plunge freezing method[6–8], developed four decades ago, which rapidly freezes macromolecule-containing solutions at liquid nitrogen temperature, yielding cryogenic specimens suitable for cryo-EM analysis.

Obtaining high-quality cryo-specimens is crucial for successful cryo-EM structure determination[9]. Despite the widespread adoption of cryo-EM nowadays, specimen preparation remains a major challenge, affecting the efficiency and success yield of structural analysis[10]. The conventional specimen preparation method entails blotting with filter paper[11,12], which hinders precise control over the uniformity and reproducibility of vitreous ice thickness and sample distribution[13]. Regulating the thickness of ice by adjusting blotting time and blotting force is technically challenging and relies heavily on personal experience. When dealing with new target samples, limited guidance from previous experience necessitates multiple trial-and-error experiments to determine the optimal conditions[14]. A more serious problem emerging with increasing evidence is the adsorption of macromolecules to the air–water interface (AWI) or supporting substrate within the thin solution layer, presenting a substantial challenge for high-resolution cryo-EM structural analysis[15,16]. This interfacial adsorption can lead to

[1]Ministry of Education Key Laboratory of Protein Sciences, Beijing Frontier Research Center of Biological Structures, School of Life Sciences, Tsinghua University, Beijing, China. [2]Tsinghua-Peking Joint Center for Life Sciences, Tsinghua University, Beijing, China. [3]State Key Laboratory of Precision Measurement Technology and Instruments, Department of Precision Instrument, Tsinghua University, Beijing, China. [4]These authors contributed equally: Zi Yang, Jingjin Fan. ✉e-mail: ouyang@mail.tsinghua.edu.cn; hongweiwang@tsinghua.edu.cn; zhouyuzxy@mail.tsinghua.edu.cn

denaturation and/or preferred orientation of target molecules[17,18], compromising successful structure determination[19,20]. Previous attempts to address the issue involved reducing the spot-to-plunge time[21]. However, the macromolecules inside a liquid layer thinner than 100 nm interact with the AWI at a time scale of milliseconds, still imposing a challenge for cryo-specimen preparation[22].

Electrospray ionization (ESI) is a soft ionization technique commonly used in native mass spectrometry (MS) for analyzing liquid-phase proteins[23]. ESI involves the application of high voltage to a conductive solution, generating charged droplets at atmospheric pressure, which subsequently lead to desolvated macromolecule ions[24]. Compared to other ionization methods, ESI minimizes damage to the molecules and efficiently ionizes biological macromolecules across a broad size range. The production of multiply charged protein ions by ESI facilitates the study of proteins and related polypeptide fragments in the kilo-Dalton to mega-Dalton range[25,26], particularly in mass spectrometers with $m/z$ ranges extending into the thousands. Moreover, nano-ESI offers the advantage of requiring only a minimal solution volume (nl) for quantitative analysis[27]. While ESI was previously used for cryo-specimen preparation, technological constraints limited the attainment of high-contrast protein sample micrographs and high-resolution 3D structures[28].

In this study, we developed a cryo-EM specimen preparation method ESI-cryoPrep using ESI to deposit small droplets of macromolecule solution on EM grids, followed by fast-freezing at liquid nitrogen temperature. The technique uses an electric field to create charged droplets, which undergo fission during flight, followed by the deposition of macromolecule-containing droplets onto the supporting substrate. The use of voltage-assisted spray technology eliminates the requirement for sample blotting with filter paper, and enhances the controllability and repeatability of the cryo-specimen preparation process. Notably, this method simultaneously addresses the challenges of macromolecule adsorption to the AWI and the supporting substrate in cryo-specimen. We anticipate that this hybrid approach will have implications not only for cryo-EM sample preparation but also for the integration of MS and cryo-EM techniques, opening new possibilities for innovative applications.

## Results

### Design of the ESI-coupled experimental setup

ESI-cryoPrep uses an ESI device to generate liquid droplets on EM grids for cryo-EM specimen preparation (Fig. 1a). The conventional ESI tip, with an outer diameter of approximately 200 µm, features a coaxial configuration consisting of an inner fused capillary for sample delivery and an outer metal capillary for nebulizing gas delivery. A positive high voltage is applied to the metal capillary to produce charged droplets. A metal EM grid covered with reduced graphene oxide (rGO) film[29] serves as the supporting substrate as well as the counter (negative) electrode for spray. The application of a high electric potential difference leads to the formation of a Taylor cone[30] at the spray tip, generating charged droplets with or without the assistance of nebulizing gas[31]. During the flight of the droplets toward the EM grid, the droplets undergo solvent evaporation and droplet fission on charges exceeding the Rayleigh stability limit[32,33] (Fig. 1b).

An optimal condition can be achieved by adjusting parameters such as sample solution injection flow rate, spray voltage, sample concentration and nebulizing gas flow rate. To mitigate the impact of high electrolyte concentration and prevent crystallization during desolvation, NaCl salt in the buffer solution is replaced with volatile ammonium acetate salt. Real-time monitoring of temperature and humidity during droplet formation is conducted using installed sensors. The droplet deposition process is observed using a high-frame-rate CCD camera (Fig. 1c,d). Subsequently, the droplets collected on the EM grid are vitrified through manual plunging in liquid ethane cooled by liquid nitrogen for cryo-EM analysis.

It is commonly asserted that charges on an electrospray (ES) droplet are evenly spaced across its surface to minimize electrostatic potential energy, stabilized by Coulomb repulsion forces[34]. Previous research has proposed the formation of an electrochemical equilibrium charge layer at the liquid–gas interface, as described by the Poisson–Boltzmann equation[35]. This layer, with a thickness on the order of Debye's length, may potentially repel protonated proteins from the liquid surface, thereby preserving high-resolution features for structural analysis.

### Optimization of ESI conditions for specimen preparation

We optimized the conditions for specimen preparation by adjusting various parameters for ESI, mainly including solution injection flow rate, spray voltage, sample concentration and landing distance from the capillary tip to the grid. These parameters are crucial in maintaining the native status of macromolecules during droplet formation. The introduction of nebulizing gas also influences the preservation of native structures. To assess the spraying performance, we examined the percentage of intact proteins under different parameters using apo-ferritin as a testing model by negative staining EM and native MS analysis (Extended Data Figs. 1 and 2).

To control the size of initial droplets generated from the ESI tip, we varied the solution injection flow rate at a level of several hundred nl min$^{-1}$, comparable with what is typically used for liquid chromatography with MS analysis. Intact apo-ferritin particles, characterized by a circular shape with a central hole in their conformation, were considered suitable for subsequent cryo-EM analysis. A reduction in the flow rate to less than 100 nl min$^{-1}$ exhibited a noticeable decrease in the percentage of intact proteins (Extended Data Fig. 1a,b). Furthermore, MS analysis revealed charge state shifts in the protein ions, signifying the unfolding of macromolecules (Extended Data Fig. 2a). The optimized flow rate range was determined to be 100 to 300 nl min$^{-1}$, striking a balance between particle preservation and ice layer thickness. We also observed that proteins are more prone to denaturing at higher spray voltage and lower target protein concentrations.

We screened various spray voltages to achieve a steady droplet mist while minimizing disruption to the protein structure. Although higher voltages facilitate smoother spraying, they pose a higher risk of electric discharge and damage to the protein structure. We found that a spray voltage of 3 kV is optimal for preserving intact proteins while maintaining a steady spray. Adjusting the distance between the ESI tip and the EM grid allowed us to control ionization progression and minimize the excessive loss on the collecting plane. The distance between the ESI tip and the receiving end was 1–1.5 cm. With a sampling time of tens of seconds, the accumulated volume of liquid droplets on the grid surface was controlled within the range of 25–50 nl. Severe impairment, indicated by the appearance of collapsed or scattered dots in raw micrographs, was observed with the introduction of $N_2$ gas flow, indicating the relatively detrimental effects of an increased degree of desolvation. The percentage of well-shaped particles markedly decreased with increased gas flow rates. At higher gas flow rates, charge states for 24-subunit particles shifted to higher values with the detected peaks of globular proteins substantially broader, and the signal for two-subunit particles became more prominent, all indicating severe damage to the proteins.

Nano-ESI was also explored as an alternative solution for comparison. It uses borosilicate glass capillaries with a pulled spray tip (1–4 µm) to generate charged droplets with sizes less than 100 nm, 100–1,000 times smaller than those by the conventional ESI. We compared the performance of gold coating or metal (stainless steel or platinum) electrodes for applying the high spray voltage and observed that platinum electrode preserved the highest proportion of intact proteins in both EM and native MS analyses (Extended Data Figs. 1a,b and 2f). This is likely due to the chemical inertness of platinum. By contrast, stainless steel is more prone to causing electrochemical reactions on proteins, therefore damaging protein structures severely.

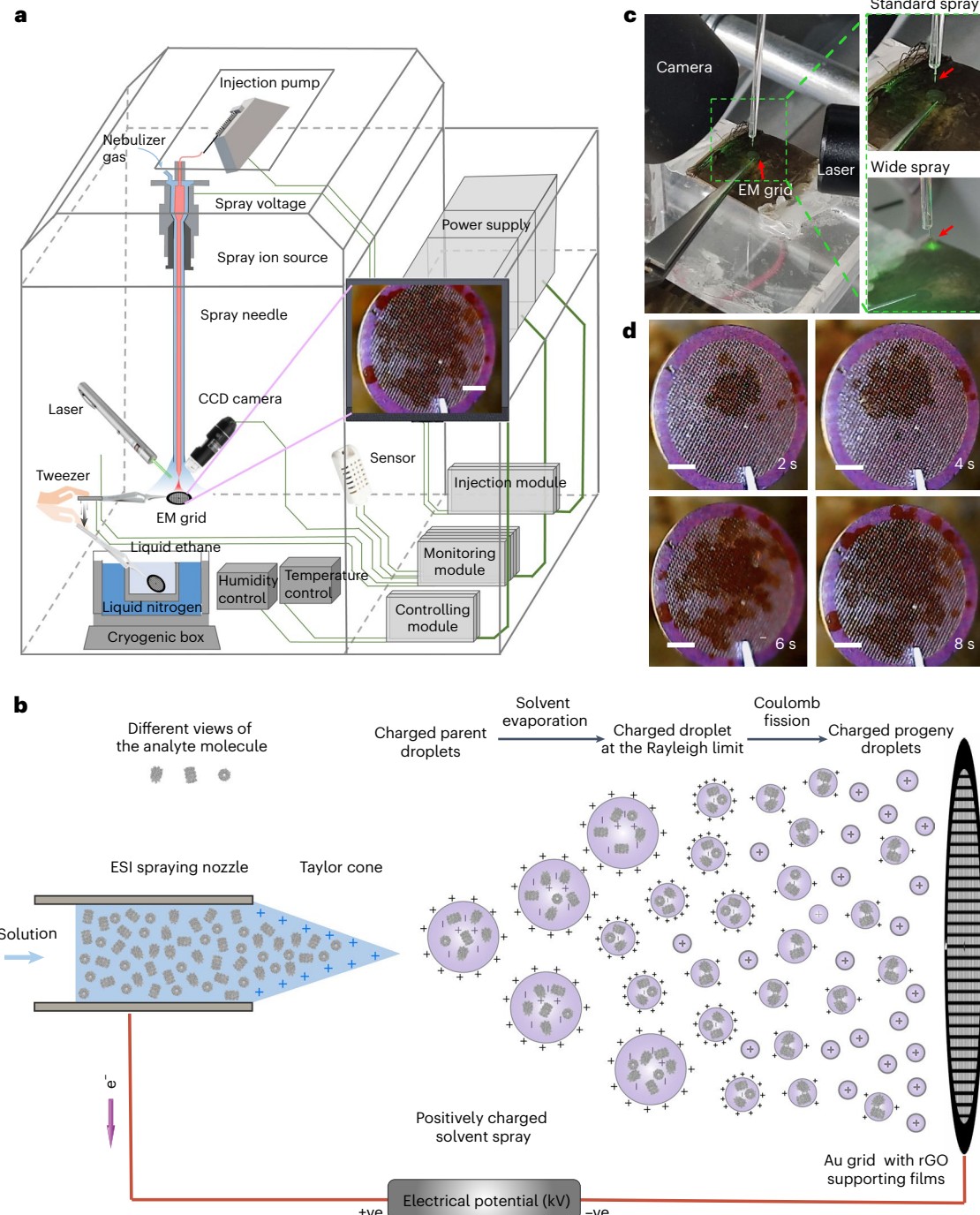

**Fig. 1 | Schematic representation of ESI-cryoPrep design and device.**
**a**, Arrangements of ESI-cryoPrep physical device. Device schematics, with an inset zooming in on the grid surface depicting deposited droplets. **b**, Mechanisms of ESI. **c**, Close-ups of the sprayer tip, grid, laser and spray. Zoomed camera and laser inspection of grid surface showing deposition of sprayed droplets. **d**, Sequential snapshots of the spray process were captured at 2, 4, 6 and 8 s. Scale bars, 500 µm.

## Cryo-EM analysis of ESI-cryoPrep specimens

The evaluation process mentioned above by negative staining EM and native MS provided valuable insights into the key parameters for cryo-EM specimen preparation. The spraying conditions were modified to ensure intact protein samples and the attainment of an optimal thickness of the vitreous ice layer. Subsequently, we assessed the efficacy of the ESI-cryoPrep method for high-resolution cryo-EM analysis using five model macromolecules: 70S ribosome, 20S proteasome, apo-ferritin, human angiotensin-converting enzyme 2 (ACE2) and streptavidin encompassing a wide range of molecular weights and biochemical properties.

Initial examination of these cryo-EM specimens under transmission electron microscope at different magnifications allowed for the assessment of droplet distribution and vitreous ice thickness (Fig. 2a–c). All four samples, 70S ribosome, 20S proteasome, apo-ferritin and ACE2, exhibited very thick areas in the center of the grid montage (Fig. 2a). It was observed that the suitable ice thickness for data acquisition displayed a gradient of contrast away from the dark zones on the grid atlas, while the contrast remained relatively consistent within the same square for most holes (Fig. 2b). High-magnification micrographs with thin ice and fast Fourier transforms revealed distinct particle

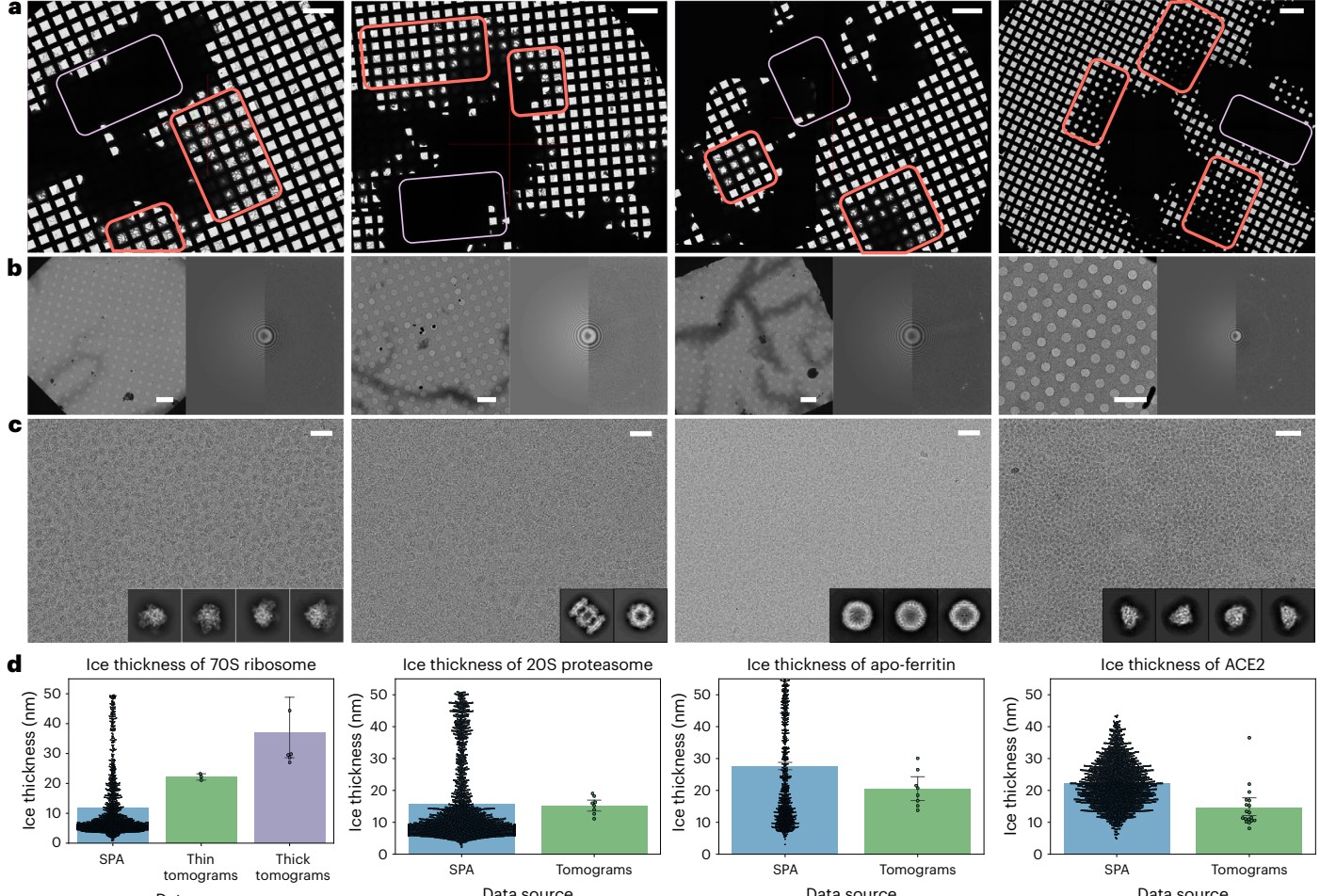

**Fig. 2 | Demonstration of ice thickness of specimens prepared with ESI-cryoPrep. a–c**, Micrographs of four cryo-specimens, 70S ribosome, 20S proteasome, apo-ferritin and ACE2, were screened at low (×60 or ×130) (**a**), medium (×320 or ×740) (**b**) and imaged at high (×29,000 or ×105,000) (**c**) magnifications under optimal conditions. Orange squares indicate areas with suitable contrast for data collection, while purple squares indicate areas with opaque contrast due to thick ice. The ice thickness distribution is highest in the middle and decreases in a wedge shape toward the edges, providing suitable thin ice for macromolecules of various sizes. Micrograph fast Fourier transforms (**b**) reveal thin ice thickness, robust particle signals and distinct graphene diffraction patterns. **d**, Ice thickness measurements were obtained from 3D particle distributions in both single-particle datasets and tomograms for the four tested proteins. Ice thickness in the single-particle data is on the same scale as the monolayer particle distribution in the tomograms. $n = 1,426$, 3 and 6, $n = 1,720$ and 8, $n = 1,038$ and 8, $n = 1,768$ and 18 independent micrographs and tomograms were analyzed for each macromolecule specimen. The central measure of the error bar is the mean, while the error bar represents the 95% confidence interval. Scale bars, 200 μm (**a**), 40 μm (**b**) and 40 nm (**c**).

features even at low defocus values (Fig. 2c). In areas with optimal ice thickness, cryo-specimens demonstrated particle distribution in a single layer with relatively uniform ice thickness, as confirmed by both single-particle analysis (SPA) and 3D cryo-electron tomography (cryo-ET) analysis (Fig. 2d). The particle distribution ranged from 15–30 nm, aligning favorably with the diameter of the proteins (10 to 26 nm), indicating an optimal ice thickness that minimizes background noise in the vitreous ice.

We acquired cryo-EM micrographs of the specimens and performed image processing for the five types of macromolecule, resulting in 3D reconstructions of the 70S ribosome, 20S proteasome, apo-ferritin, ACE2 and streptavidin at resolutions of 2.7, 2.0, 2.1, 3.3 and 1.9 Å, respectively (Fig. 3a–e, top). When performing 3D reconstruction at comparable particle levels, the ESI-cryoPrep method demonstrated notably superior resolution in the 3D density maps compared to the control (Extended Data Fig. 3). The estimated *B* factors of the five specimens were determined as 50, 62, 68, 100 and 52 Å², respectively. The cryo-EM 3D density maps exhibited sufficient quality for unambiguous atomic modeling (Fig. 3a–e, bottom). Notably, in the regions of the highest resolution, fine features such as holes in tyrosine benzene rings

and extended signals of side chains of arginine, isoleucine, valine and serine were clearly resolved. The successful reconstruction of these specimens demonstrated the effectiveness of the ESI-cryoPrep method in preserving the natural structure of macromolecules.

**Particle distribution in specimens prepared by ESI-cryoPrep**

To evaluate the particle distribution in ESI-cryoPrep specimens, we analyzed and compared the above datasets that yielded successful cryo-EM reconstructions with those obtained from conventional cryo-specimen preparation. ESI-cryoPrep specimens exhibited a higher proportion of particles contributing to high-resolution reconstruction compared to the conventionally prepared specimens (Extended Data Fig. 4). Previous studies have indicated that protein particles tend to adsorb at the AWI[17] and potentially at the supporting film interface[15,20], which can lead to damage and/or preferential orientations of macromolecule particles, compromising high-resolution SPA reconstruction. To investigate the influence of interfaces (both air–liquid interface and solid–liquid interface) on particle distribution, we compared the particle angular orientation of the five macromolecule specimens prepared using ESI-cryoPrep and conventional methods. The angular orientation heatmaps of the

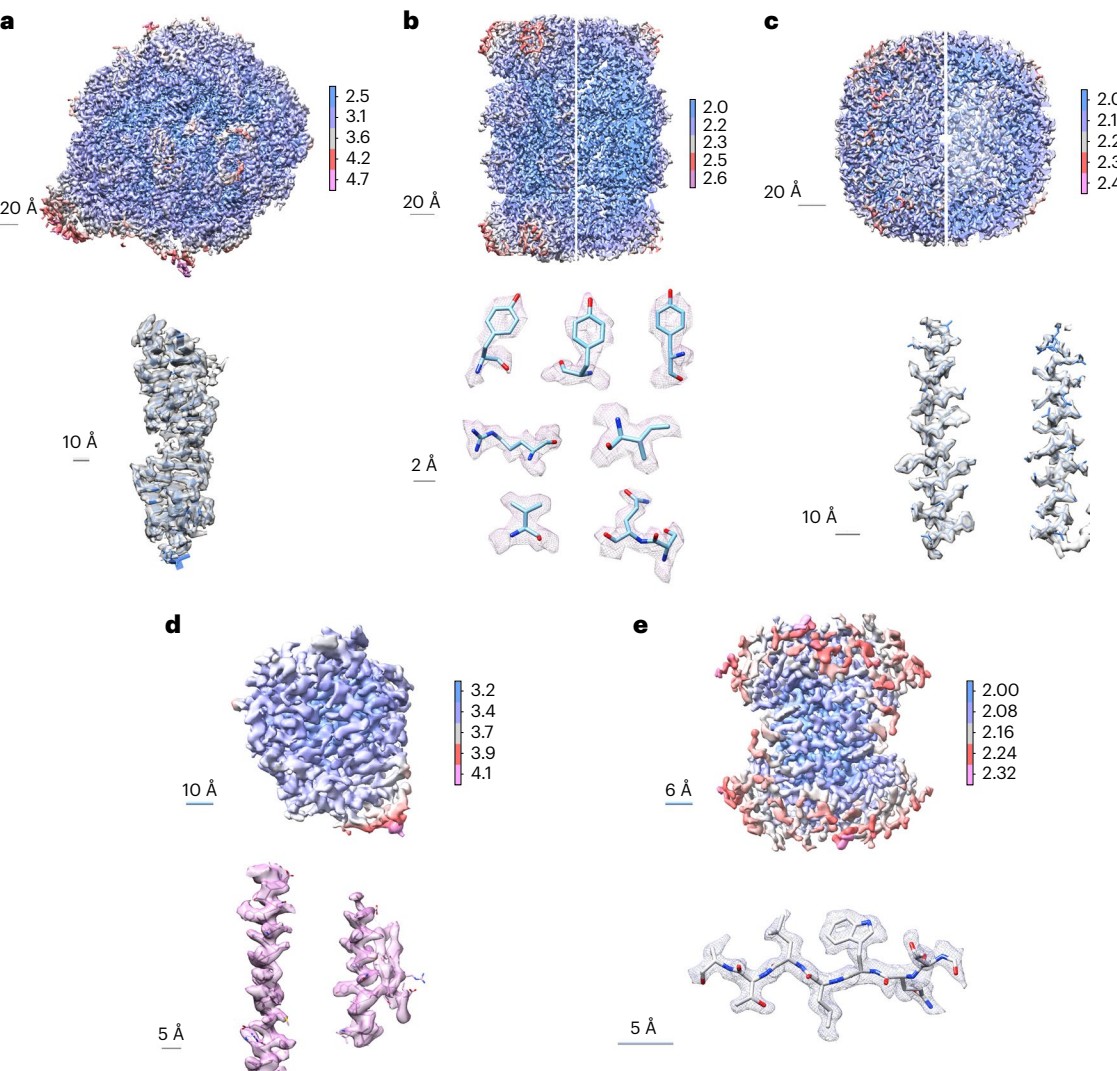

**Fig. 3 | Cryo-EM reconstructions of the five macromolecules by ESI-cryoPrep.**
**a**–**e**, 3D reconstructions of the 70S ribosome (**a**), 20S proteasome (**b**), apo-ferritin (**c**), ACE2 (**d**) and streptavidin (**e**) with atomic resolutions of 2.7, 2.0, 2.1, 3.3 and 1.9 Å, respectively (top). The views from the outside (left) and middle sections (right) of 20S proteasome and apo-ferritin have been combined to reveal structure credibility. Zoomed regions of the five macromolecule structures show well-defined densities of side chains, carbonyl bonds and benzene rings (bottom).

five macromolecule types (70S ribosome with C1 symmetry, 20S proteasome with D7 symmetry, apo-ferritin with O symmetry, ACE2 with C1 symmetry and streptavidin with D2 symmetry) demonstrated that ESI-cryoPrep specimens notably enhanced angular projection coverage and increased sampling rates of Fourier components across most viewing directions. This is in stark contrast to the angular orientations of the same macromolecules prepared using the conventional method on supporting films (Fig. 5d and Extended Data Fig. 5).

The observed angular distribution prompted further investigation of particle distribution near the air–liquid and solid–liquid interfaces in the ESI-cryoPrep specimens. Spatial arrangement analysis of particles was conducted using cryo-ET, where multiple tomograms were collected for each macromolecule specimen. Through tilt-series alignment, automated particle selection and subtomogram averaging, the averaged structure was mapped back to the tomograms with determined *xyz* coordinates and calculated orientation information. Previous studies have shown that particles in conventional methods with supporting films were distributed on both interfaces[36]. However, in the ESI-cryoPrep specimens, particles were observed to be arranged within the middle of the thin ice and not directly attached to either interface. Figure 4a illustrates that in each representative tomogram, particles

from the four test samples were distributed in a thin monoplane, consistent across all collected tomograms. They were arranged more condensed in the middle of the vitreous ice. The *zx* slice of tomograms for the four types of macromolecule prepared by ESI-cryoPrep revealed that particles distributed slightly closer to the graphene side but not attached to the AWI. The spacing between the lower edge of particles and the graphene layer is estimated to be approximately 4–6 slices, corresponding to a range of approximately 2–3 nm (Extended Data Figs. 6 and 7). A representative result demonstrated that apo-ferritin particles packed predominantly in a monoplane and fully embedded within the vitreous ice (Fig. 4b).

To obtain more comprehensive statistical results regarding particle distribution in the specimen, we used per-particle defocus values derived from refining contrast transfer function (CTF) parameters during SPA. This precise calculation of particle height within the ice layer was achieved through the determination of per-particle defocus values, refined based on phase residuals. The alignment of variations in particle defocus with their respective *z* heights in vitreous ice was instrumental in our analysis. We used least-squares plane fitting[37] to extrapolate particle distribution patterns and gauge ice thickness across the entire SPA dataset, which contributed to high-resolution 3D reconstruction.

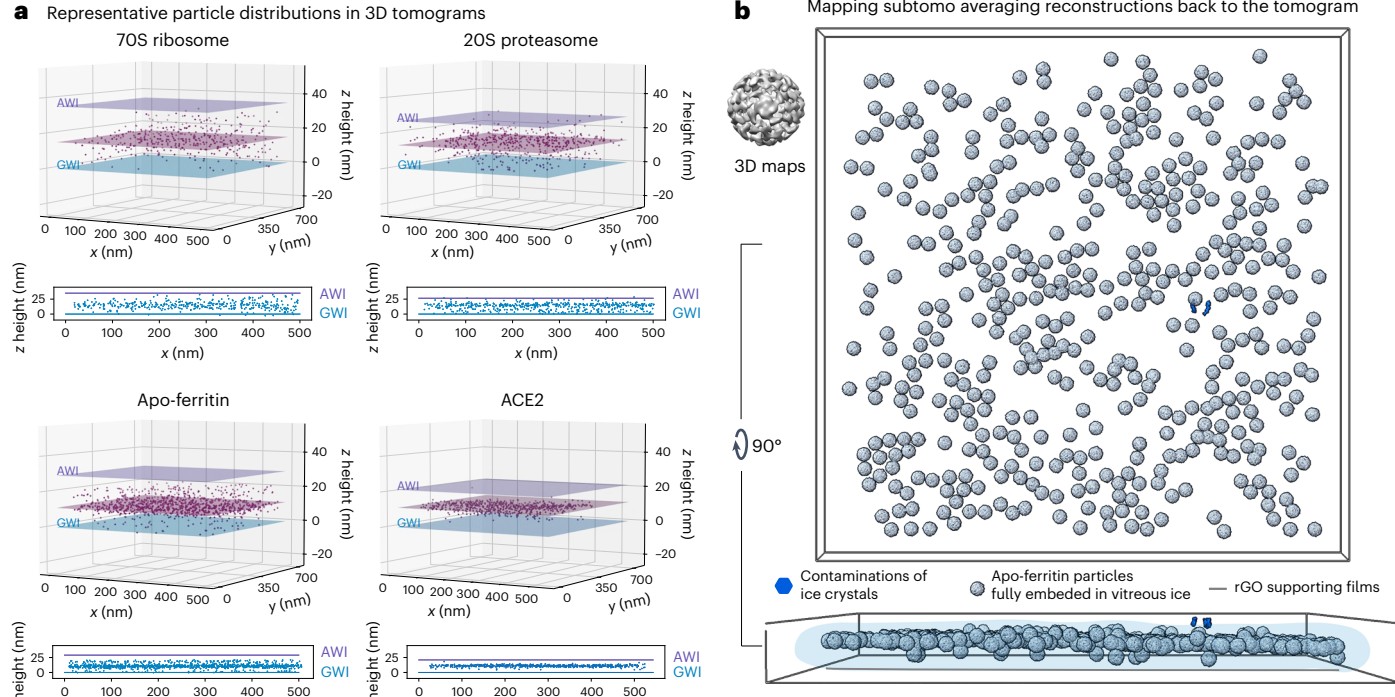

**a** Representative particle distributions in 3D tomograms

**b** Mapping subtomo averaging reconstructions back to the tomogram

**Fig. 4 | Tomographic determination of the spatial distribution of protein particles prepared with the ESI-cryoPrep method. a**, Tomographic analysis of the spatial distribution of the four types of macromolecule in representative holes was performed separately. Particle distributions in 3D tomograms indicate that most particles of 70S ribosome, 20S proteasome, apo-ferritin and ACE2 are distributed in monolayers within a thin layer of vitreous ice. The spatial arrangement of particles in *x*, *y* and *z* coordinates is shown with fitting planes of particles (magenta), AWI (purple) and GWI (blue) in the 3D perspective. In the 2D perspective, the arrangement of particles in the *zx* plane is shown with the same annotations for AWI (purple) and GWI (blue). **b**, Visualization of apo-ferritin particles in a representative tomogram. The averaged subtomogram (subtomo) reconstruction is mapped to the aligned tomogram with determined *xyz* coordinates and orientation information. Silver spheres, blue blocks and gray lines indicate apo-ferritin particles, ice crystal contamination and the layer of supporting films, respectively. The transparent light blue shape indicates the silhouette of vitrified buffer, determined by ice crystals and contaminations on the supporting films.

This analysis was applied to datasets from ESI-cryoPrep specimens as well as specimens prepared using the conventional method on graphene-based supporting films or in suspending ice.

As a negative control for particle distribution in suspending ice, the apo-ferritin dataset (EMPIAR-10424) was analyzed in a similar way. The *z* heights of apo-ferritin particles were depicted in barplots and subsequently fitted using the kernel density estimation (KDE) method. The *z* height plots from two distinct micrographs of the apo-ferritin sample (Fig. 5a, purple) demonstrated nonuniform distributions with inverted symmetry and high absolute skewness values. Each plot exhibited a steep and gentle slope on either side of the peak, indicating particle adsorption at both AWIs. The intense distribution and steep slope corresponded to densely adsorbed protein complexes, while the long tail suggested freely scattering particles distributed away from the interfaces. The distributions also demonstrate the deviations of the mode from the mean and the large absolute value of the skewness.

A similar analysis was performed for apo-ferritin, ACE2 and streptavidin on supporting films prepared using the conventional preparation method, displaying characteristics consistent with particles adhering to either the AWI or graphene–water interface (GWI) (Fig. 5a, green). The distribution with a higher proportion of skewed sides, signifying dominant adsorption, hints at potential adsorption at the air or graphene interfaces. Notably, the bimodal distributions of the *z* height of apo-ferritin particles on rGO grids suggested protein adsorption at both the AWI and GWI.

The same analysis was applied to the ESI-cryoPrep specimens, where the KDE fits of the datasets were characterized by relatively symmetric (Gaussian-like) peaks, with only a small fraction of extreme values (Fig. 5a, blue). This observation strongly indicates that particles

in the ESI-cryoPrep specimens are predominantly embedded within the central region of the vitreous ice, showing minimal adsorption at the AWI or GWI.

To ensure an unbiased representation of micrographs, the entire dataset was considered for statistical analysis. In probability theory and mathematical statistics, moments of a function serve as quantitative measures characterizing the shape of a set of points (*X*) within that function. Skewness, the third central moment, quantitatively measures the degree of asymmetry in a distribution. Negative skewness commonly denotes a left-skewed distribution, while positive skewness indicates a right-skewed distribution in the case of a unimodal distribution. A skewness value of zero suggests a symmetric distribution, where the mean, median and mode coincide, and gains or losses on each side exhibit equal frequency. As a guideline, skewness values less than −1 or greater than 1 indicate a highly skewed distribution, values between −1 and −0.5 or between 0.5 and 1 suggest moderate skewness, and values between −0.5 and 0.5 indicate an approximately symmetric distribution.

Skewness distributions were examined for control data (apo-ferritin in suspending ice, apo-ferritin, ACE2 and streptavidin with supporting films) and ESI-cryoPrep data of 70S ribosome, 20S proteasome, apo-ferritin, ACE2 and streptavidin, respectively (Fig. 5b). Notably, skewed distributions were observed for particles on UltraAu-Foil gold grids (apo-ferritin data, EMPIAR-10424) and the conventional method with supporting films, displaying more pronounced asymmetry on either side compared to ESI-cryoPrep data. In contrast, the distribution of ESI-cryoPrep data appeared more symmetrical. In ESI-cryoPrep data, the positive *z* heights of the 70S ribosome, 20S proteasome and ACE2 correspond to the AWIs, while the positive *z* heights of the apo-ferritin and streptavidin correspond to the GWIs.

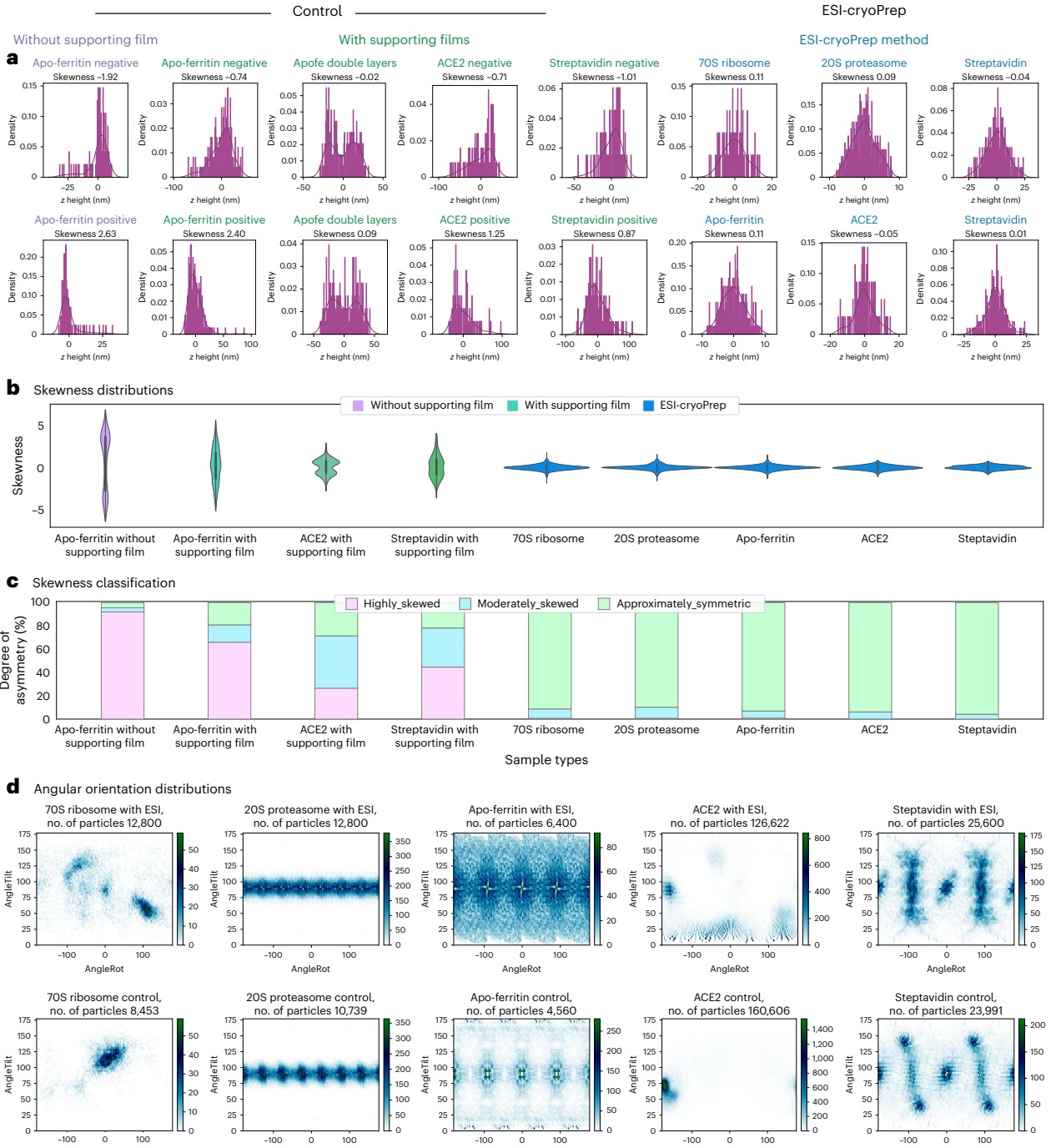

**Fig. 5 | Spatial distribution and angular orientation analysis with single-particle data. a**, $z$ height distribution of particles in a single micrograph. $z$ height is the distance of particles from the fitted plane, which corresponds to spatial distributions along the $z$ axis. The peak of the KDE fitting curve indicates the mode of $z$ heights, while the zero point corresponds to the mean distance. The unimodal distributions of $z$ heights of apo-ferritin, ACE2 and streptavidin particles are distributed in asymmetrical shapes with a long tail when supported on the holey gold grid with two interfaces in contact with the air (purple), or the EM grids with one interface in contact with the air and the other in contact with the supporting film (green), respectively. In some micrographs, $z$ heights of apo-ferritin particles prepared on rGO show bimodal distributions, indicating adsorption at both AWIs and GWIs. 70S ribosome, 20S proteasome, apo-ferritin, ACE2 and streptavidin particles (blue) show Gaussian-like symmetrical distributions of $z$ heights at the main peak when prepared using the ESI-cryoPrep method. These distributions appear to have the same mean, mode and median, and low absolute skewness value. **b**, Skewness distribution of particles' $z$ height across the entire SPA dataset. **c**, Skewness classification of particles' $z$ height

across the entire SPA dataset. The control particle $z$ height distributions (apo-ferritin without or with supporting films, ACE2 and streptavidin with supporting films from conventional methods) tend to be asymmetric with larger absolute skewness values, while ESI-cryoPrep distribution appears to be more symmetric with close-to-zero skewness values. The color code is compatible with that in **a**. For each sample skewness distribution, $n$ = 2,821, 3,275, 2,209, 280, 1,606, 1,801, 1,168, 1,793 and 827 independent micrographs were analyzed, progressing from left to right. The center dot of the inner boxplot represents the median, while the box's limits denote data values within the interquartile range (IQR). Whiskers extend from the box to the furthest data point within 1.5 times the IQR. Data points beyond the whiskers are identified as outliers and are excluded from the primary figure. **d**, Heatmap of angular orientations of the five protein samples, 70S ribosome (first panel), 20S proteasome (second panel), apo-ferritin (third panel), ACE2 (fourth panel) and streptavidin (fifth panel). Data using the ESI-cryoPrep method are shown on the top panel, while the control data on supporting films using the conventional method are shown on the bottom. AngleRot, AngleTilt are two given Euler angles calculated in RELION.

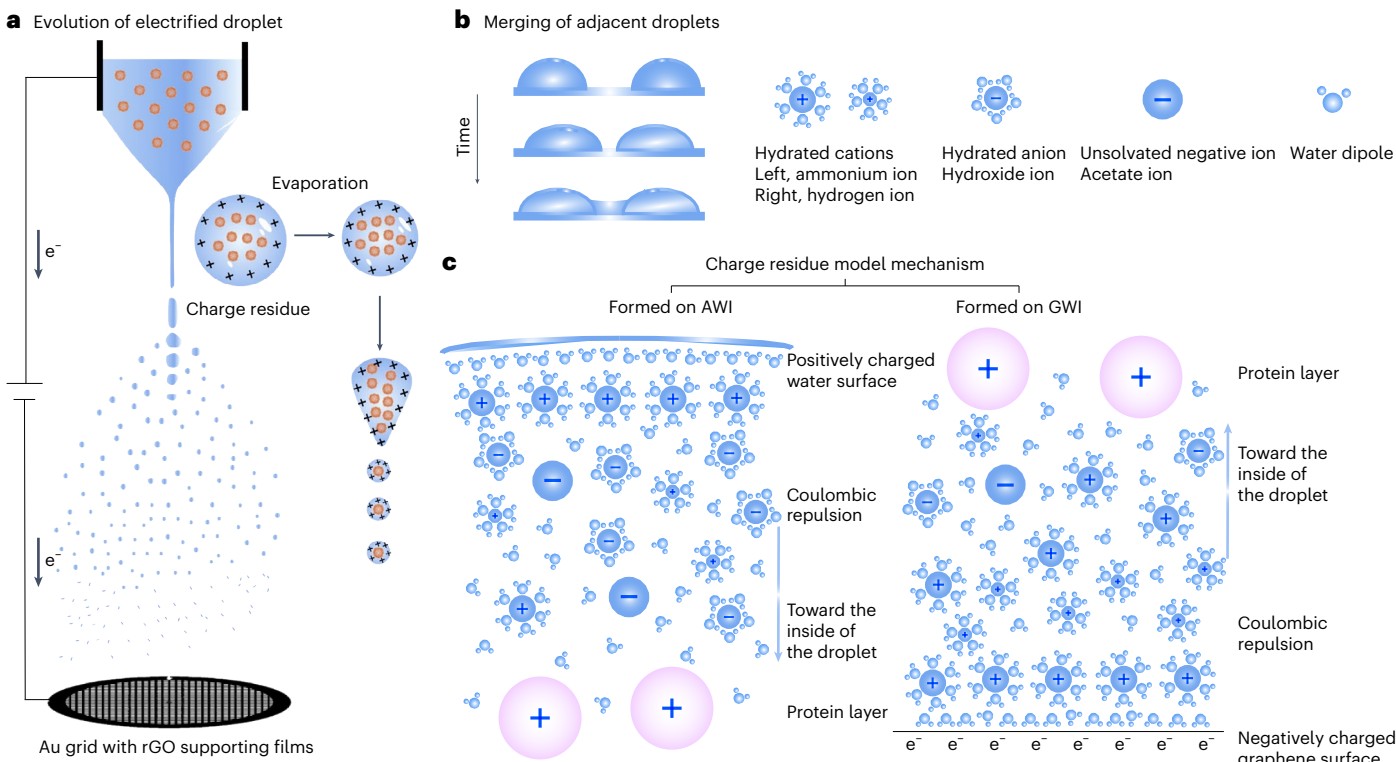

**Fig. 6 | Mechanism of spreading and wetting of spray droplets, and the charge residue model at interfaces. a**, Schematic representation of the evolution of electrified droplets in the ESI-cryoPrep equipment. **b**, Adjacent droplets form a water bridge and merge into a larger droplet through wetting effects. As droplets merge, their surface area decreases, resulting in a reduction of the total surface energy. **c**, The charge residue model at the AWI or GWI serves as a barrier, preventing direct interactions between protein particles and the opposing phase. Polar water solvents exhibit robust interactions with the charged droplet surface or the charged electrode. The radial charge gradient arises from mutual Coulombic repulsion among excess charges within the charge residue model of droplets. The top panel features legends for different ions.

The statistical results support the notion that the $z$ heights of most particles in the ESI-cryoPrep dataset were distributed more symmetrically (Fig. 5c), consistent with results from the single micrograph data.

### Working model for the ESI-cryoPrep method

The present cryo-sample preparation technique using ESI involves the generation of charged microdroplets from a protein solution[38]. ESI operates at atmospheric pressure where the analyte solution is infused into a metal capillary held at a positive electric potential of several thousand volts. The solution forms a Taylor cone at the capillary tip, emitting fine mist droplets when the Raleigh limit is reached. These initial droplets, with micrometer-scale radii, carry an excess of positive charge primarily from protonated ions[39,40]. The charge density experiences a pronounced increase during solvent evaporation as they migrate toward the supporting substrate (Fig. 6a). It is established in the ESI process that small ions are inclined to accumulate at the surface of charged droplets, eventually transforming into dry ions through ion evaporation[39,41]. By contrast, large, multiply charged and highly hydrophilic protein ions reside at the center, becoming dry ions after the evaporation of water and small ions. This phenomenon is known as the charge residue model[42,43]. In the ESI-cryoPrep method, the charge residue model mechanism aids in distributing protein ions away from both the AWI and GWI on landing on the supporting substrate, ultimately preserving their native forms.

On impact with a stationary solid surface, droplets undergo deformation influenced by inertial force, viscous force and surface tension. Various droplet characteristics, including size, velocity and surface morphology, affect their behavior on collision, such as deposition, spreading, bouncing or splattering[44]. In the case of electrospray, droplets of varying sizes and velocities undergo a complex collision process

with the substrate. Previous research demonstrated the measurement of kinetic energy gain in charged particles moving within an electric field[45]. The collision process involves the conversion of kinetic energy and potential energy before collision into kinetic energy, potential energy and work done by friction after collision, following the law of conservation of energy. Droplets spread on collision, experiencing a substantial increase in surface potential energy, consuming most of their initial kinetic energy, while another portion of the kinetic energy dissipates as frictional work. Assuming an initially spherical droplet with a near-zero incident angle, on impact with the substrate, the droplet spreads bilaterally, creating a thicker central region and thinner regions at the sides.

Adjacent droplets can merge, forming a water bridge through wetting effects (Fig. 6b). The exposed surface of a spherical droplet encounters inward forces, leading to contraction until it reaches a limit that is determined by the surface tension. When two droplets merge, the reduction in surface area decreases total surface energy, releasing the remaining energy. Electrostatic attraction between positively charged droplets and the negatively charged grid surface overcomes surface tension, facilitating further spreading and wetting of the droplets.

As demonstrated above, the ESI-cryoPrep approach proficiently confines macromolecules within the core of vitreous ice, preventing the adherence of target molecules to the AWI or GWI. This phenomenon is presumably attributed to uneven ion distributions in terms of particle sizes and charges[46–48], akin to mechanisms discussed earlier for ion evaporation and charge residual models in the ESI process. Merged droplets on the substrate retain excessive charges, preventing protein ions from reaching AWI and substrate–water interfaces, as depicted in Fig. 6c.

## Discussion

In ESI of MS, analytes transit from the condensed phase to the gas phase and undergo ionization. Previous studies assumed that the charged water droplets produced in the final Coulomb fission event, containing a neutral macromolecule, would be slightly larger than the macromolecule. Consequently, during the final solvent evaporation, all droplet charges would be transferred to the macromolecule's surface. In our setup, the droplets containing macromolecules are not fully evaporated into the gas phase, therefore the molecules stay fully immersed in the liquid, with much less protonation in comparison with their gaseous ionic states. As demonstrated above, the relatively less protonated macromolecules maintain their high-resolution native structures in the small droplets produced by ESI. This actually provides evidence of the native preservation of the macromolecules in ESI-MS.

As proposed in our hypothetical model to explain electrochemical reactions and their effects on protein structures under different conditions, electrostatic forces and the conversion of kinetic energy to surface potential generated an ice layer with a thickness gradient, advantageous for accommodating protein complexes of various sizes. Euler angle distributions for the five macromolecule specimens demonstrated markedly improved angular projection coverage in Fourier space. The nature of excessively charged droplets in ESI-cryoPrep shielded protein particles from interfacial effects, ensuring their full embedding within the ice layer without adhesion to interfaces. Unlike recent inventions relying on self-wicking nano-wire grids, ESI-cryoPrep uses regular EM grids with conductive supporting films, enabling additional investigations using tilt scheme or cryo-ET.

Cryo-EM captures multiple macromolecular conformational states, offering opportunities for time-space-resolved studies. The ESI-cryoPrep technique, with precise instrument tuning, can potentially study fast molecular events in a time-resolved manner. Time-resolved samples can be prepared using conventional methods or a dual-jet system for mixing reactants on the EM mesh. The ESI-cryoPrep method offers additional potential benefits in high-resolution cryo-EM analysis, preparing specimens for biochemical reactions and coupling MS analysis with cryo-EM.

The nanospray needle in nano-ESI, with an outer diameter in the order of tens of nanometers, generates nanoscale initial droplets with a high desolvation tendency, facilitating the production of extremely small final droplets for the analysis of ultrathin samples. However, nano-ESI encounters challenges due to notable electrochemical effects under current conditions, causing particle dissociation and potential vaporization of the thin fluid film. The potential of nano-ESI under softer electrospray conditions is yet to be determined.

We anticipate that the relatively low cost and ease of maintenance make the ESI-cryoPrep suitable for broader application, with potential expansion to various laboratories and research institutions. Future exploration will include diverse protein samples, such as membrane proteins and tiny crystals, advancing cryo-EM sample preparation.

## Online content

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

## Methods

### ESI-cryoPrep specimen preparation

The 70S ribosomes and streptavidin were purchased from the NEB company (catalog nos. P0763S and N7021S), while *Thermoplasma acidophilum* 20S proteasomes[49], human heavy-chain apo-ferritins[50] and human ACE2 (ref. 51) were prepared as previously described, respectively. Optimal conditions for ESI-cryoPrep cryogenic specimen preparation were adopted in the following experiments. The ESI needle was connected to the positive electrode of a d.c. power supply (+3 kV), and 20 µl macromolecule solution (diluted with 100 mM ammonium acetate solutions to a concentration of about 0.5–4 µM) was injected using a precise and stable automatic injection device (Supplementary Table 1). The negative electrode was connected to the tweezer holding the metal grid (covered with a layer of rGO as supporting films), which had previously been subjected to glow discharge at a medium current for 12 s in a Harrick Plasma instrument (Supplementary Table 2). At a moderate injection flow rate (100–300 nl min$^{-1}$) and a sampling duration of tens of seconds, we maintained the cumulative volume of liquid droplets on the grid surface within the range of 25–50 nl (Supplementary Fig. 1). The grids were then either rapidly stained with droplets of 3% uranyl acetate for negative staining EM analysis or manual frozen in nitrogen-cooled liquid ethane for cryo-EM analysis.

### Negative-staining electron microscopy

The prepared negatively stained grids of apo-ferritin were screened on a Tecnai F20 Twin microscope (FEI) operating at 200 kV, using a 4,000 CCD camera (Gatan, Inc.) at a nominal magnification of ×50,000 with a calibrated pixel size of 1.7 Å per pixel. Two-dimensional (2D) class averages were initially generated by manually picking particles from the binned raw micrographs, which were then used as templates for automatic particle picking using template matching in RELION. Picked particles were then extracted and a subset of particles was selected from the 2D classification output. The proportion of intact particles was calculated by dividing the number of selected particles after 2D classification by the number of particles picked from template matching.

### Single-particle cryo-EM data collection and preprocessing

The 70S ribosome, 20S proteasome and apo-ferritin dataset were collected on a Titan Krios microscope (FEI) equipped with a Gatan K3 Summit camera operating at 300 kV. The ACE2 and streptavidin datasets were collected on a Titan Krios microscope (FEI) equipped with a GIF Quantum energy filter (slit width 20 eV) operating at 300 kV. Videos were recorded in super-resolution mode on the K3 Summit direct electron detector (Gatan, Inc.) with a micrograph size of 11,520 × 8,184 pixels and calibrated pixel sizes of 0.485 Å (70S ribosome, 20S proteasome and apo-ferritin), 0.4187 Å (ACE2) and 0.2595 Å (streptavidin) per super-resolution pixel. The micrographs were taken with defocus at a range of −0.7 to −1.4 µm. The exposure dose per frame was 1.5625 e$^-$/Å$^2$ therefore the total exposure dose of each video was 50 e$^-$/Å$^2$ with 32 frames. The total exposure time was set to 2.56 s at 0.08 s per frame (70S ribosome, 20S proteasome and apo-ferritin), and 1.28 s at 0.04 s per frame (ACE2) and 0.43 s at 0.0133 s per frame (streptavidin). Data acquisition and preprocessing were performed with AutoEMation[52], SerialEM[53] and the facility's homemade wrapper TsinghuaTitan.py.

### Image processing

Super-resolution videos were motion-corrected using the Motion-Cor2 program[54] with an 11 × 11 patch number, and then downsampled with Fourier cropping by a factor of 2 to a pixel size of 0.97 Å (70S ribosome, 20S proteasome and apo-ferritin), 0.8374 Å (ACE2) and 0.5191 Å (streptavidin) per counted pixel. For details, refer to Supplementary Table 3. Parameters of the micrographs' CTF were estimated by an exhaustive search with CTFFIND4.1 (ref. 55) and Gctf[56].

Micrographs with even particle distribution and uniform vitreous ice were manually screened.

Particle picking was performed by template matching in RELION-4.0 (ref. 57) and CNN-based automatic picking in Topaz[58], which incorporates a deep learning model. Particles were extracted at a binning of twofold to speed up alignment and classification. A simplified scheme for image processing procedures is presented in the Extended Data Figs. 7–10. Initial models were calculated from the 1,000-particle dataset using the stochastic gradient descent algorithm in the CryoSPARC[59] program package or downloaded from the Electron Microscopy Data Bank (EMDB) and low-pass filtered to 45 Å to prevent model bias. The 2D and 3D analyses were performed in RELION-3.1 (ref. 60), RELION-4.0 (ref. 57), RELION-5.0 (ref. 61) and CryoSPARC[59]. CTF and aberration refinement, and Bayesian polishing were performed as standard procedures to increase the final resolution and reduce the *B* factor value of the reconstruction error. The final resolution was determined by the cutoff of Fourier shell correlation (FSC) at 0.143 (ref. 62), and the highest resolution reconstruction, determined by the gold-standard FSC between two half-maps from two independently sampled subsets, went through multiple trials of CTF refinement and Bayesian polishing iterations. The five types of macromolecule, 70S ribosome, 20S proteasome, apo-ferritin, ACE2 and streptavidin, were reconstructed at final resolutions of 2.77, 2.04, 2.15, 3.28 and 1.95 Å, respectively. Validations of the local resolution were assessed through a wrapper in RELION and ResMap[63]. The maps colored by local resolution were rendered with UCSF Chimera and ChimeraX[64].

Crystal structures from previous studies were aligned and fitted to 3D maps in ChimeraX, then refined in PHENIX. The structures include those of the 70S ribosome (Protein Data Bank (PDB) ID 6W6P), 20S proteasome (PDB 1PMA), apo-ferritin (PDB 1FHA), ACE2 (PDB 1R42) and streptavidin (PDB 7DY0). The crystal structures fit the reconstructions well for most regions, except for the two alpha helices of ACE2, which required refinement of their relative angles to the rest of the structure. Models for the 70S ribosome, 20S proteasome, apo-ferritin and streptavidin were manually fitted using COOT[65], followed by real-space refinement with restraints of secondary structure and noncrystallographic symmetry using PHENIX[66]. The atomic model of ACE2 was generated using cryoNet (https://cryonet.ai/), a differentiable neural network-based model-building software.

### Particle distribution analysis

The Euler angle distributions of 3D reconstructions were plotted using 2D heatmaps from two given Euler angles (AngleRot, AngleTilt) calculated in RELION. The angular sampling ranges of structures with symmetry (D7 for 20S proteasome, O symmetry for apo-ferritin and D2 for streptavidin) were taken into account in the final heatmap.

Particles were plotted in 3D graphics according to *x*, *y* and *z* coordinates, showing the spatial distribution of inclined planes relative to the *xy* plane. Simply dealing with variations in per-particle defocus values failed to provide realistic ice thickness and 3D particle distribution patterns. The least-squares plane fitting was applied to fit a plane that could best characterize the spatial distribution of most particles. The histogram of the distances from particles to the fitted plane was used to detect whether the particles were distributed in single or double layers. The ice thickness was calculated from the extent of most particles (80%) in the direction orthogonal to the fitted plane. The degree of asymmetry of the KDE fit was measured through the third central moment, skewness. The higher the absolute value is, the more asymmetric the KDE fit, meaning particles are adsorbed onto the interfaces with a steeper slope and abundant amounts.

### Cryo-ET data collection and reconstruction

Cryo-ET data were acquired using the Titan Krios 300 kV electron microscope (FEI) equipped with a 20 eV slit-width energy filter (for ACE2) or without the energy filter (for 70S ribosome, 20S proteasome

and apo-ferritin) and automatedly collected with SerialEM software[53]. Tilt series were recorded in super-resolution mode at nominal magnifications of ×23,000 and ×64,000, with calibrated pixel sizes of 0.625 Å (for 70S ribosome, 20S proteasome and apo-ferritin) and 0.6795 Å (for ACE2) per super-resolution pixel, respectively. Data collection used a dose symmetric tilt scheme ranging from −40° to 40°, or −50° to 50° with a 3° base increment and defocus cycling from −4.8 to −5.2 μm, or −2.8 to −3.4 μm. Each tilt angle was recorded with eight frames (0.213 s per frame), and the total dose for each tilt series was 81 or 105 e⁻/Å². Stage shift and beam-induced motion were corrected using MotionCor2 (ref. 54), and tomograms were aligned and reconstructed in IMOD[67]. Automated 3D particle picking was performed using Dynamo[68]. The 3D reconstruction and mapping of the sample particles were refined with EMAN[69] and RELION-4.0 (ref. 57).

### Reporting summary

Further information on research design is available in the Nature Portfolio Reporting Summary linked to this article.

### Data availability

The following publicly available data were used in the paper: apo-ferritin control dataset EMPIAR-10424. Crystal structures determined in previous work were aligned and fit to 3D maps in ChimeraX and then real-space refined in PHENIX for 70S ribosome (PDB ID 6W6P), 20S proteasome (PDB 1PMA), apo-ferritin (PDB 1FHA), ACE2 (PDB 1R42) and streptavidin (PDB 7DY0). The coordinates of the ACE2 cryo-EM structure were deposited in the Protein Data Bank under accession no. PDB 8JWH. The corresponding cryo-EM map was deposited in the EMDB under accession EMD-36683. Source data are provided with this paper.

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

### Acknowledgements

We thank F. Sun and X. J. Huang (Institute of Biophysics, CAS) for their help with the apo-ferritin sample, H. Cheng (Tsinghua University) for the 20S proteasome sample, X. Q. Wang (Tsinghua University) for the ACE2 sample and N. Liu and J. Xu (Tsinghua University) for control data prepared on rGO grids. We are grateful to J. L. Lei, X. M. Li, F. Yang, N. Y. Zhou and T. Yang at the Tsinghua University Branch of the National Protein Science Facility (Beijing) for their technical support on the Cryo-EM and High-Performance Computation platforms. We acknowledge financial support from the National Natural Science Foundation of China (grant nos. 31825009 to H.-W.W. and 22227807 to Z.O.). H.-W.W. is an awardee of the XPLORER PRIZE. The funders had no role in study design, data collection and analysis, decision to publish or preparation of the manuscript.

### Author contributions

H.-W.W. and Z.O. initiated the ESI-cryoPrep concept and conceived the research. Z.Y., J.F., X.Z. and J.W. performed the experiments. X.Z. designed the original installation. Z.Y., J.F. and X.F. performed the initial data evaluation. Z.Y. and J.F. prepared cryogenic specimens. Z.Y. collected the cryo-EM data, solved the cryo-EM structures, performed model building and analyzed spatial distribution data. J.F. collected the MS. H.-W.W., Z.O., J.W. and X.Z. supervised the research. Z.Y., J.F., H.-W.W., Z.O., J.W. and X.Z. contributed to writing of the paper. All the authors read and approved the paper.

### Competing interests

The authors declare no competing interests.

### Additional information

**Extended data** is available for this paper at https://doi.org/10.1038/s41592-024-02247-0.

**Correspondence and requests for materials** should be addressed to Zheng Ouyang, Hong-Wei Wang or Xiaoyu Zhou.

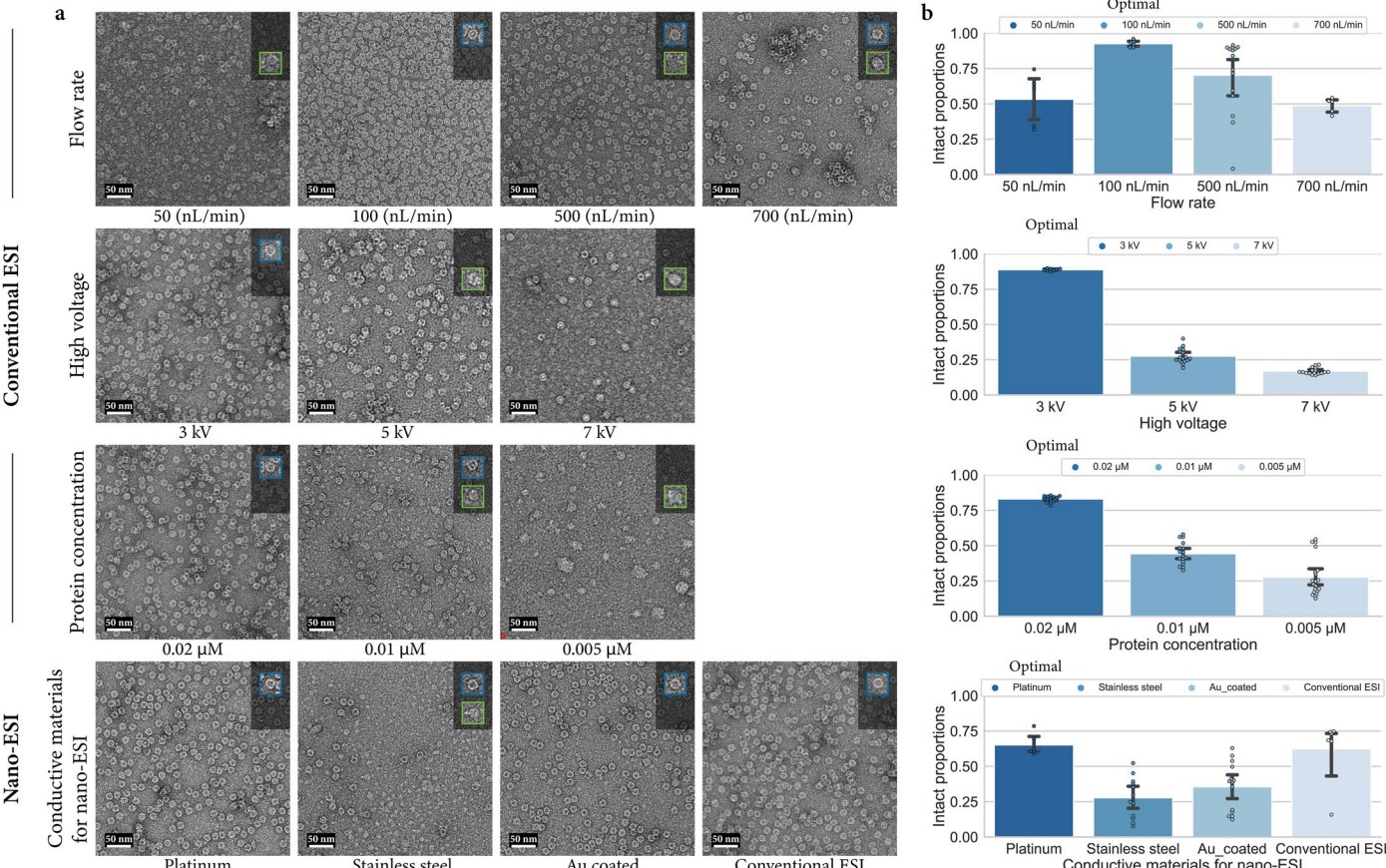

**Extended Data Fig. 1 | Parameter optimizations of sample preparation.** **a**, Structural preservation of apo-ferritin under different spraying conditions. Several parameters, like sample flow rate, high voltage, and protein concentration, affect protein samples of intact states differently. The particles in the blue squares represent the shape of the intact protein, while the particles in the green squares represent the disrupted and dispersed state. Conditions with moderately medium flow rates, lower voltages, and higher sample concentrations contribute to better structural integrity. Metal materials with low conductivity in nano-ESI would also make a difference in structural maintenance. **b**, shows the statistical analysis of the percentage of structurally intact proteins

in EM negative-staining under different spraying conditions. Micrographs were collected and analyzed independently for each flow rate, high voltage value, protein concentration, and conductive material of nano-ESI. The sample sizes (n) were 5, 6, 15, and 5 for flow rates; 8, 17, and 7 for high voltage values; 15, 17, and 20 for protein concentrations; and 6, 12, 14, and 16 for conductive materials of nano-ESI, respectively. The bars indicating the highest proportions of intact particles represent the recommended optimal parameters. The central measure of the error bar is the mean, while the error bar represents the 95% confidence interval. The source data is included in a dedicated Source Data file.

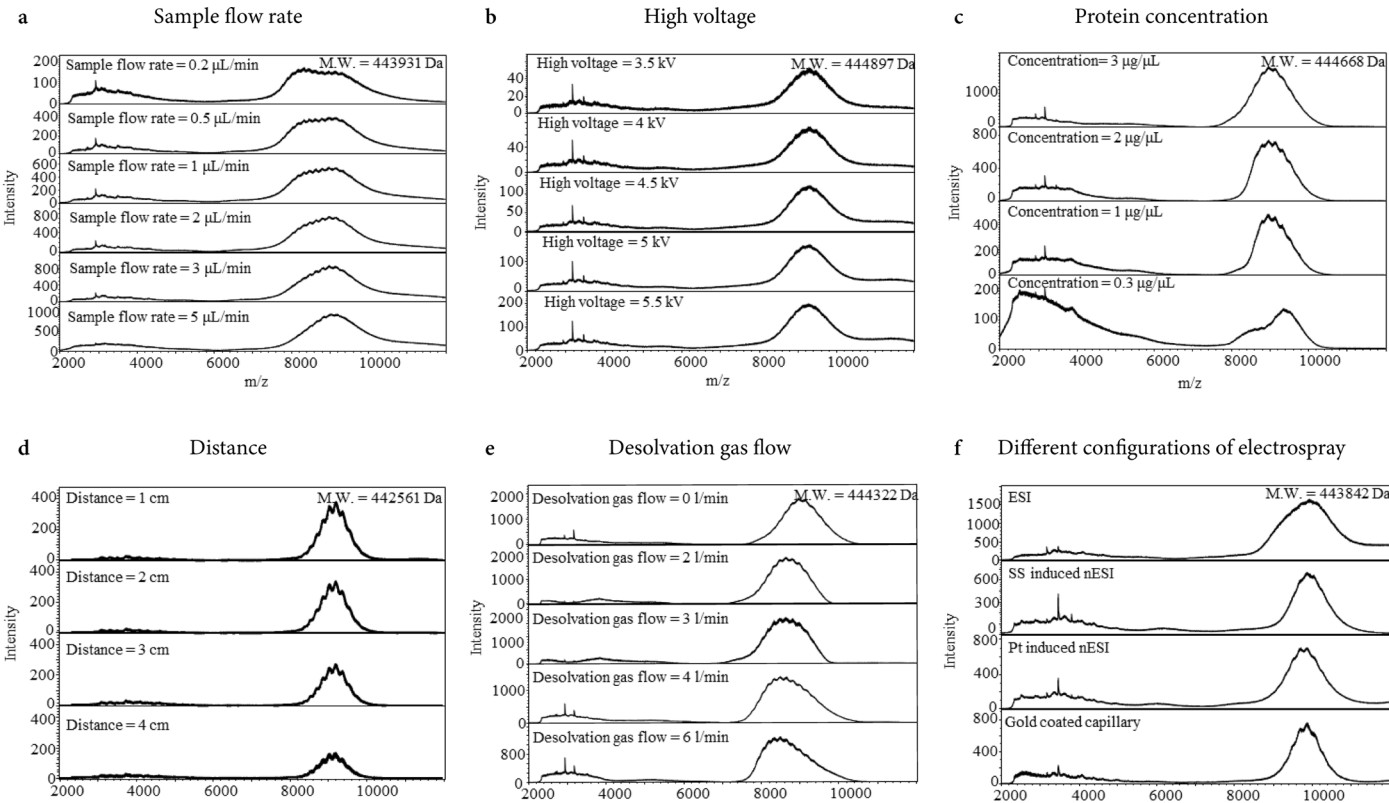

**Extended Data Fig. 2 | Mass spectrums at different conditions.** Mass spectrometry data is obtained using apo-ferritin. The favorable conditions identified in EM align with those observed in native MS analysis. From top to bottom, mass spectrums of various conditions: the injection flow rate (**a**), the high voltage (**b**), the protein concentration (**c**), the distance from the tip to the EM grid (**d**), the desolvation gas flow rate (**e**), and various conductive materials for nano-ESI (**f**). The source data is included in a dedicated Source Data file.

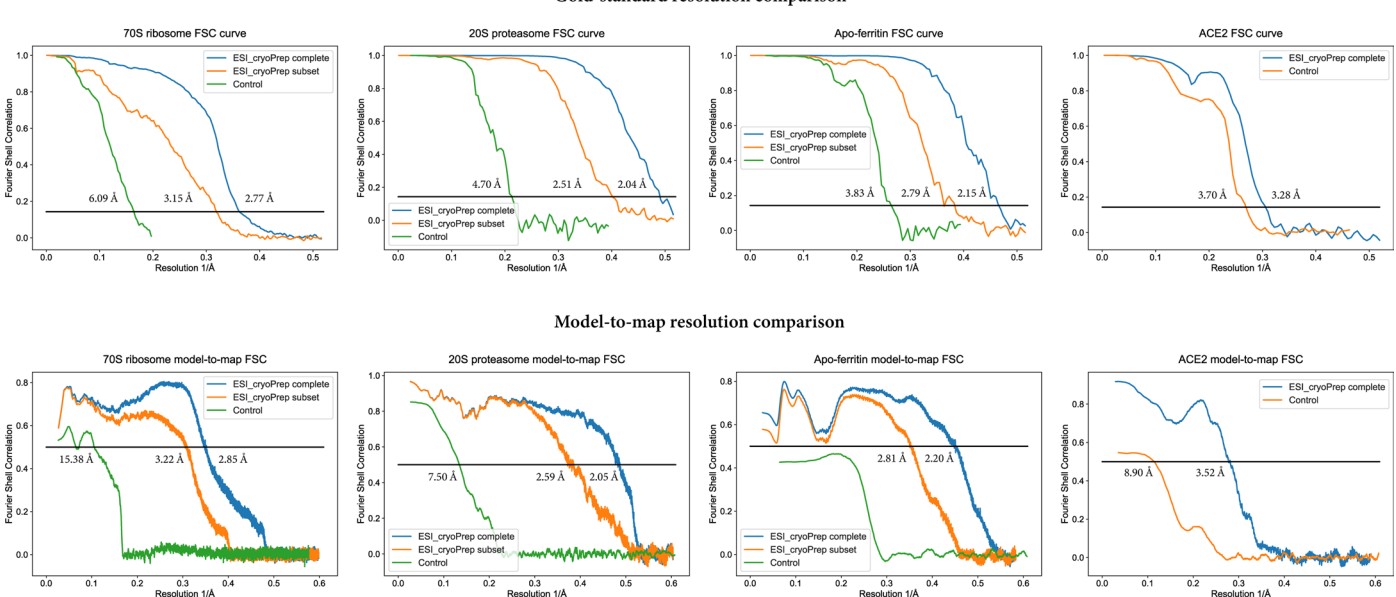

**Extended Data Fig. 3 | Gold-standard and model-to-map FSC of ESI-cryoPrep and control datasets with matched particle counts.** The particle counts of ESI-cryoPrep subsets and control data align with those presented in Fig. 5d. The plot features the graph legend in one of its corners.

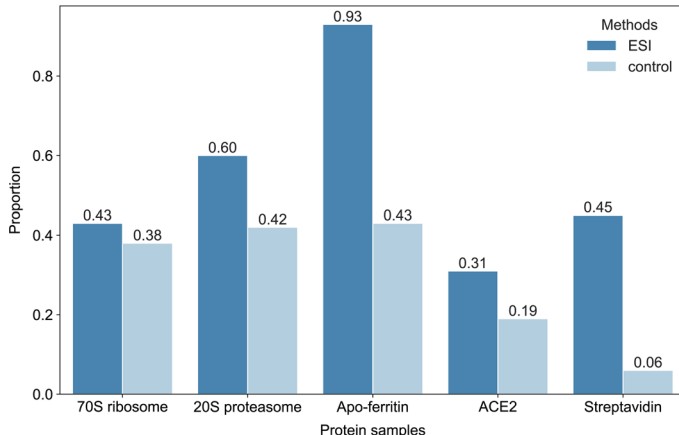

**Extended Data Fig. 4 | Particle contribution statistics for high-resolution reconstruction.** The proportion of particles contributed to high-resolution reconstruction. Particles of 70S ribosome and ACE2 using the conventional method on supporting films experienced considerable preferential orientation issues, resulting in the failure to reconstruct satisfactory 3D maps.

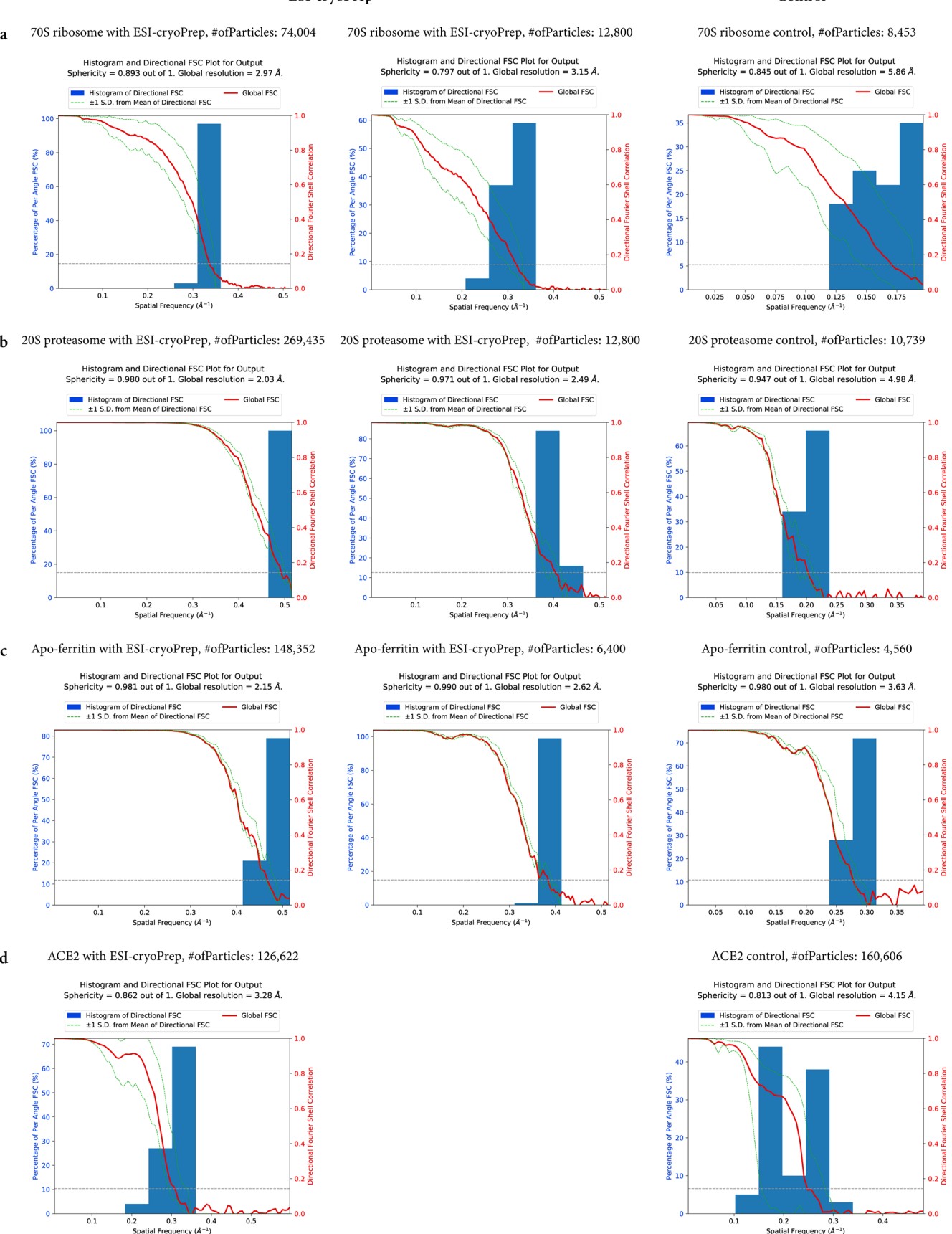

**Extended Data Fig. 5 | 3DFSC analysis of ESI-cryoPrep and control datasets with complete and matched particle counts.** 3DFSC analysis of 70S ribosome (**a**), 20S proteasome (**b**), apo-ferritin (**c**), and ACE2 (**d**). Owing to their high symmetry, the 20S proteasome and apo-ferritin exhibit slight improvements compared to the control data. By contrast, the 70S ribosome and ACE2 prepared by ESI-cryoPrep demonstrate substantial improvement of 3DFSC in all Fourier directions compared to the control.

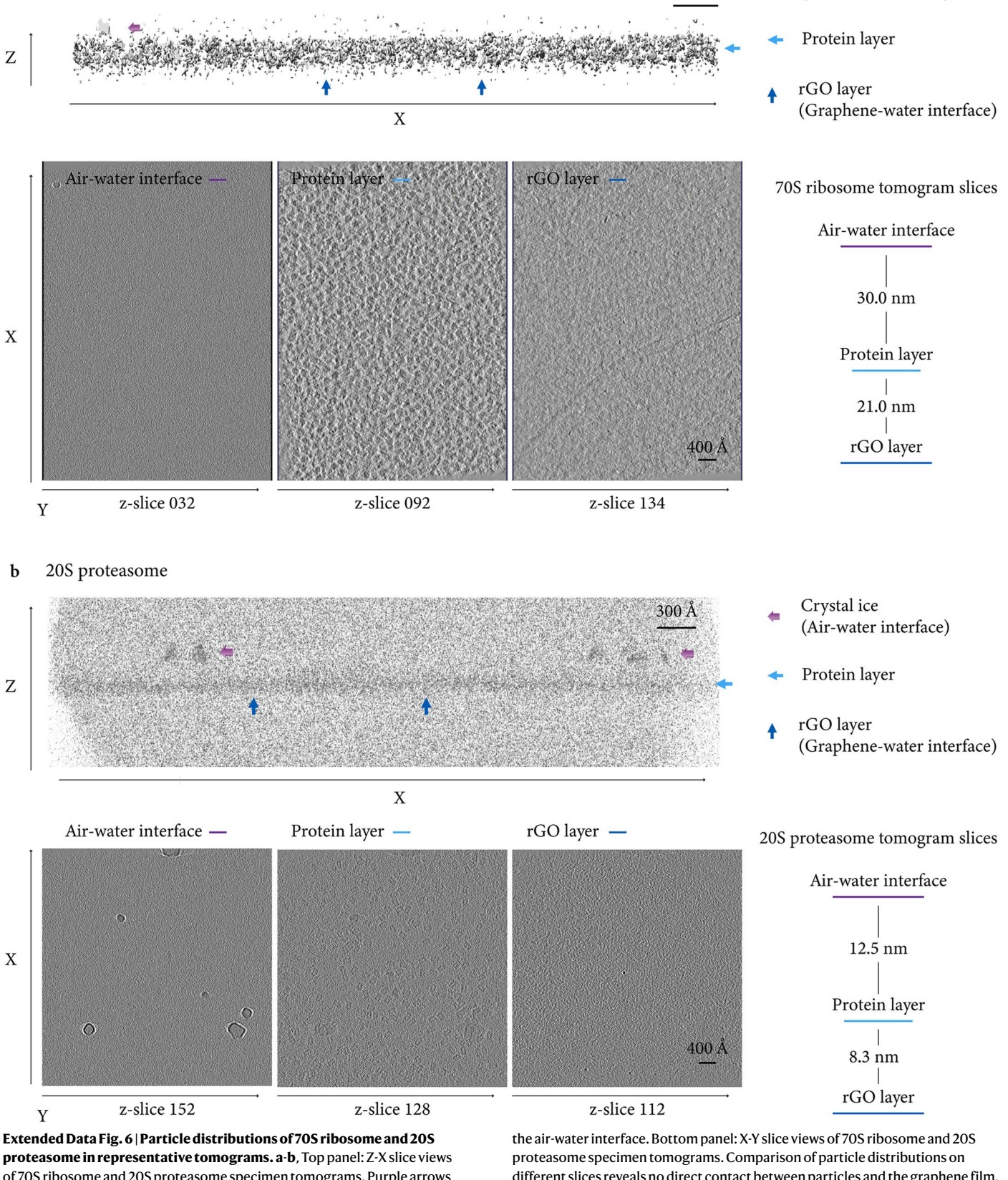

**Extended Data Fig. 6 | Particle distributions of 70S ribosome and 20S proteasome in representative tomograms. a-b**, Top panel: Z-X slice views of 70S ribosome and 20S proteasome specimen tomograms. Purple arrows indicate crystal ice, while light and dark blue arrows denote the protein layer and graphene layer, respectively, illustrating particle distribution away from the air-water interface. Bottom panel: X-Y slice views of 70S ribosome and 20S proteasome specimen tomograms. Comparison of particle distributions on different slices reveals no direct contact between particles and the graphene film. The color code corresponds to that in the top panel.

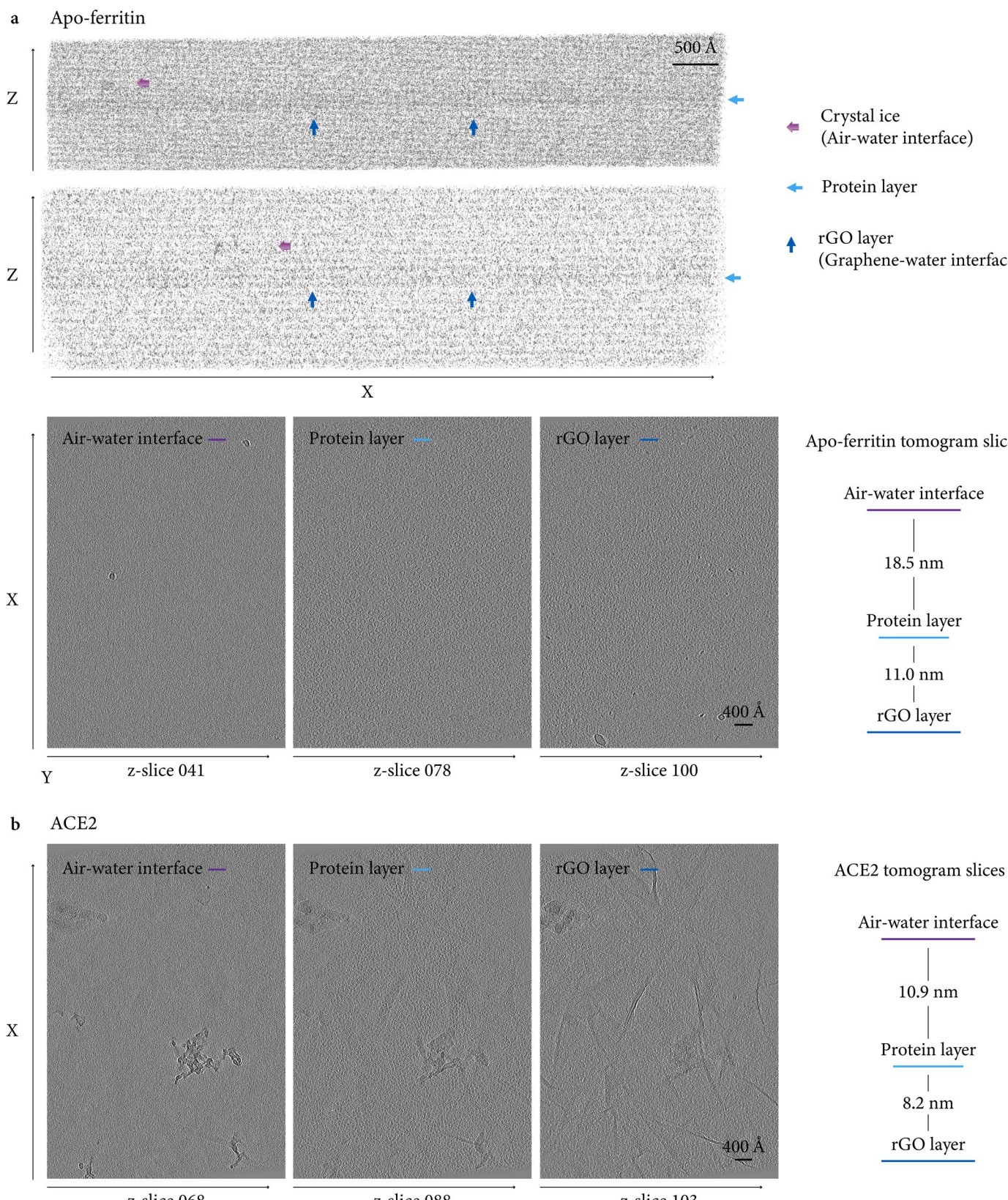

**a** Apo-ferritin

Crystal ice
(Air-water interface)

Protein layer

rGO layer
(Graphene-water interface)

Air-water interface —   Protein layer —   rGO layer —   Apo-ferritin tomogram slices

Air-water interface

18.5 nm

Protein layer

11.0 nm

rGO layer

z-slice 041   z-slice 078   z-slice 100

**b** ACE2

Air-water interface —   Protein layer —   rGO layer —   ACE2 tomogram slices

Air-water interface

10.9 nm

Protein layer

8.2 nm

rGO layer

z-slice 068   z-slice 088   z-slice 103

**Extended Data Fig. 7 | Particle distributions of apo-ferritin and ACE2 in representative tomograms. a-b**, Top panel: Z-X slice views of apo-ferritin and ACE2 specimen tomograms. Purple arrows indicate crystal ice, while light and dark blue arrows denote the protein layer and graphene layer, respectively, illustrating particle distribution away from the air-water interface. Bottom panel:

X-Y slice views of apo-ferritin and ACE2 specimen tomograms. Comparison of particle distributions on different slices reveals no direct contact between particles and the graphene film. The color code corresponds to that in the top panel.

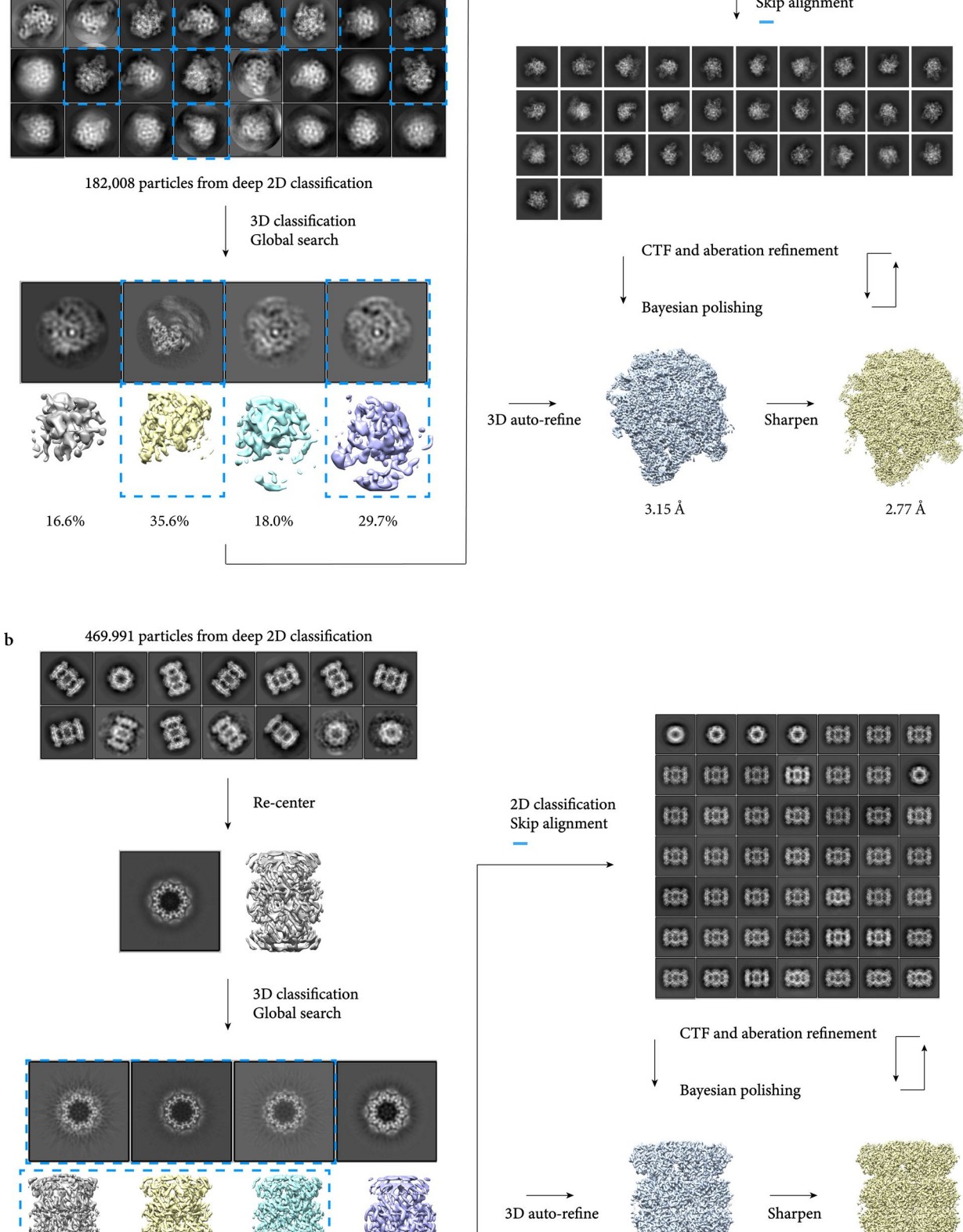

**Extended Data Fig. 8 | EM data processing flowchart for 70S ribosome and 20S proteasome.** Structure determination and analysis of 70S ribosome (**a**) and 20S proteasome (**b**).

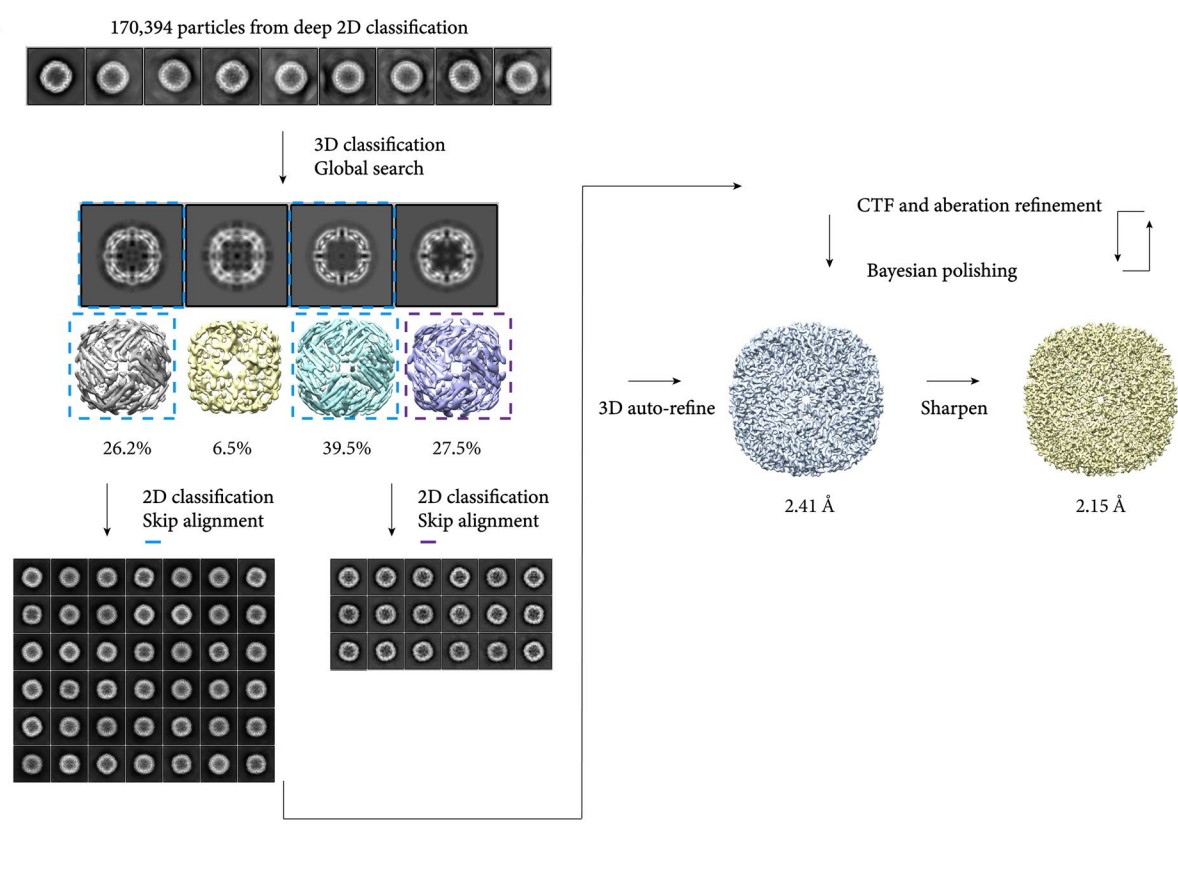

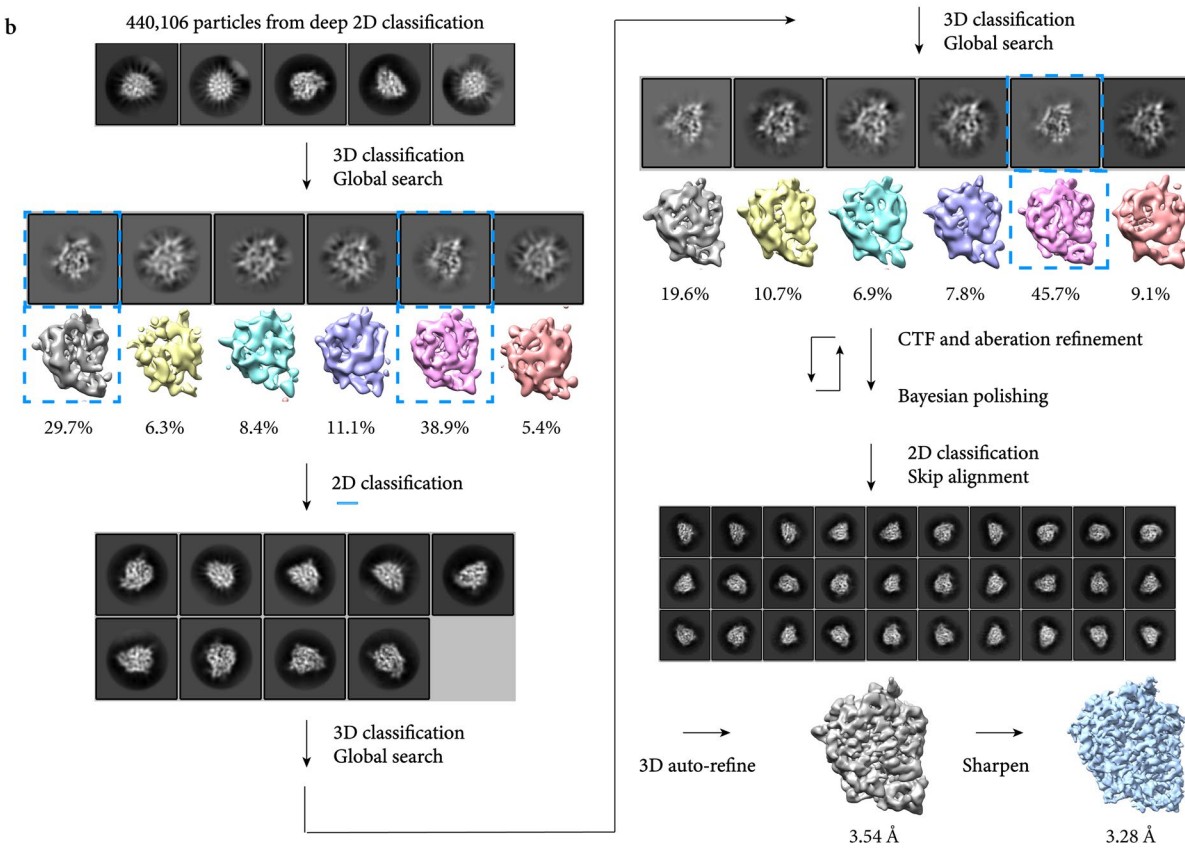

**Extended Data Fig. 9 | EM data processing flowchart for apo-ferritin and ACE2.** Structure determination and analysis of apo-ferritin (**a**) and ACE2 (**b**).

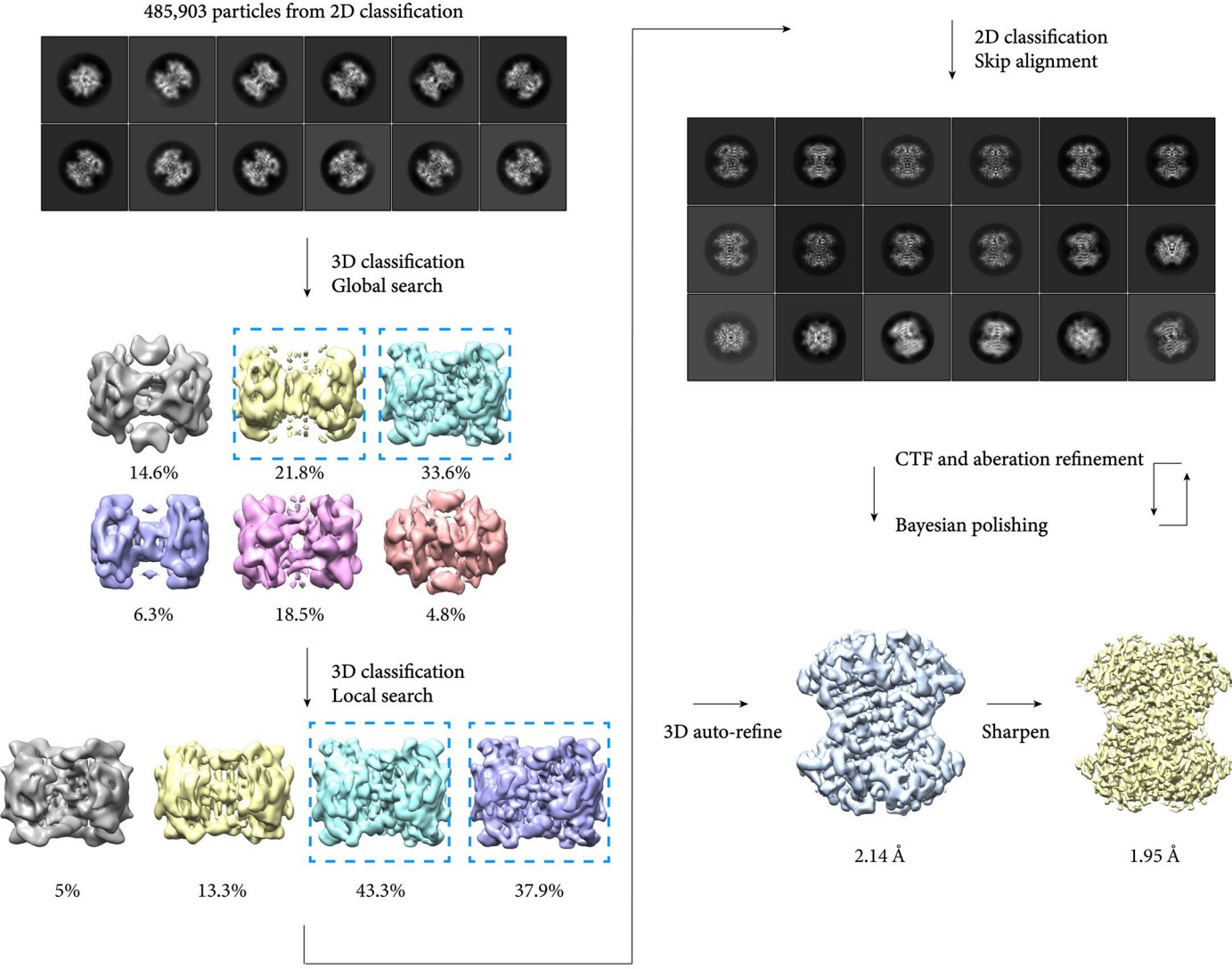

**Extended Data Fig. 10 | EM data processing flowchart for streptavidin.** Structure determination and analysis of streptavidin.

# Reporting Summary

## Statistics

For all statistical analyses, confirm that the following items are present in the figure legend, table legend, main text, or Methods section.

| n/a | Confirmed | |
|---|---|---|
| ☐ | ☒ | The exact sample size (*n*) for each experimental group/condition, given as a discrete number and unit of measurement |
| ☐ | ☒ | A statement on whether measurements were taken from distinct samples or whether the same sample was measured repeatedly |
| ☒ | ☐ | The statistical test(s) used AND whether they are one- or two-sided *Only common tests should be described solely by name; describe more complex techniques in the Methods section.* |
| ☒ | ☐ | A description of all covariates tested |
| ☐ | ☒ | A description of any assumptions or corrections, such as tests of normality and adjustment for multiple comparisons |
| ☐ | ☒ | A full description of the statistical parameters including central tendency (e.g. means) or other basic estimates (e.g. regression coefficient) AND variation (e.g. standard deviation) or associated estimates of uncertainty (e.g. confidence intervals) |
| ☒ | ☐ | For null hypothesis testing, the test statistic (e.g. *F*, *t*, *r*) with confidence intervals, effect sizes, degrees of freedom and *P* value noted *Give P values as exact values whenever suitable.* |
| ☒ | ☐ | For Bayesian analysis, information on the choice of priors and Markov chain Monte Carlo settings |
| ☐ | ☒ | For hierarchical and complex designs, identification of the appropriate level for tests and full reporting of outcomes |
| ☒ | ☐ | Estimates of effect sizes (e.g. Cohen's *d*, Pearson's *r*), indicating how they were calculated |

*Our web collection on statistics for biologists contains articles on many of the points above.*

## Software and code

Policy information about availability of computer code

| Data collection | Cryo-specimens were prepared with ESI-cryoPrep method. Single-particle cryo-EM datasets were collected using AutoEMation2.0 software written by Dr. Jianlin Lei at Tsinghua University, and SerialEM (version 3.8 and 4.0). Dose symmetric tilt-series were collected with SerialEM (version 3.8 and 4.0) software. |
|---|---|
| Data analysis | Pre-processing of SPA and tomography dataset were performed to facilitate micrograph-selecting with the facility's homemade wrapper TsinghuaSoftwares.py and a homemade script Pre-process-tomo.sh incorporating MotionCor2 and Gctf, respectively. The beam-induced motion and radiation damage of cryo-EM micrographs were corrected with MotionCor2 (version 1.6.3). Particle picking was performed by template matching in RELION (version 3.1, 4.0 and 5.0) and CNN-based automatic picking in Topaz (version 0.2.4). The 2D and 3D analyses were performed in RELION (version 3.1, 4.0 and 5.0) and cryoSPARC (4.2 and 4.4). The CTF values of motion-corrected micrographs were determined by CTFFIND4 (version 4.1.14) and Gctf (version 1.18). The structural analysis was performed in UCSF Chimera (version 1.17) and UCSF Chimera X (version 1.6). Validations of the local resolution were assessed through a wrapper in RELION (version 3.1) and ResMap (version 1.1.4). The coordinates were real-space refined in PHENIX (version 1.19) and adjusted in Coot (version 0.9.8). For cryo-ET analysis, tilt-series were aligned using IMOD (version 4.11) and EMAN2 (version 2.29) to reconstruct the 3D tomograms. Particle-picking and 3D reconstruction were analyzed using EMAN2 (version 2.29) and Dynamo (version 1.1). All these softwares are open-source. |

For manuscripts utilizing custom algorithms or software that are central to the research but not yet described in published literature, software must be made available to editors and reviewers. We strongly encourage code deposition in a community repository (e.g. GitHub). See the Nature Portfolio guidelines for submitting code & software for further information.

# Data

Policy information about availability of data

All manuscripts must include a data availability statement. This statement should provide the following information, where applicable:

- Accession codes, unique identifiers, or web links for publicly available datasets
- A description of any restrictions on data availability
- For clinical datasets or third party data, please ensure that the statement adheres to our policy

The following publicly available data were used in the manuscript: apo-ferritin control dateset EMPIAR-10424. Crystal structures determined in previous work were aligned and fit to 3D maps in ChimeraX and then real-space refined in PHENIX for 70S ribosome (PDB IDs-6W6P), 20S proteasome (PDB IDs-1PMA), apo-ferritin (PDB IDs-1FHA), ACE2 (PDB IDs-1R42), and streptavidin (PDB IDs-7dy0). All necessary data for evaluating the conclusions are provided in the paper and/or Extended Data Materials. The coordinates of the ACE2 cryo-EM structure were deposited in the Protein Data Bank under accessions 8JWH. The corresponding cryo-EM map was deposited in the Electron Microscopy Data Bank (EMDB) under accessions EMD-36683.

# Human research participants

Policy information about studies involving human research participants and Sex and Gender in Research.

| | |
|---|---|
| Reporting on sex and gender | not applicable |
| Population characteristics | not applicable |
| Recruitment | not applicable |
| Ethics oversight | not applicable |

Note that full information on the approval of the study protocol must also be provided in the manuscript.

# Field-specific reporting

Please select the one below that is the best fit for your research. If you are not sure, read the appropriate sections before making your selection.

☒ Life sciences    ☐ Behavioural & social sciences    ☐ Ecological, evolutionary & environmental sciences

For a reference copy of the document with all sections, see nature.com/documents/nr-reporting-summary-flat.pdf

# Life sciences study design

All studies must disclose on these points even when the disclosure is negative.

| | |
|---|---|
| Sample size | For TEM analysis, each condition was characterized with the number of micrographs more than 5, and tens and hundreds of particles from each micrograph were auto-detected to calculate the percentage of intact proteins. Sample sizes were determined according to the occurrence frequency in each condition state, with confidence interval error bars added.<br>For cryo-EM single particle analysis, five different biological specimens were imaged and analyzed, which are 70S ribosome, 20S proteasome, apo-ferritin, ACE2, and streptavidin. The five sample proteins, 70S ribosome, 20S proteasome, apo-ferritin, ACE2, and streptavidin, were reconstructed at final resolutions of 2.77 Å, 2.04 Å, 2.15 Å,  3.28 Å and 1.95 Å, estimated by the Fourier Shell Correction (FSC)=0.143 cutoff criteria with a number of 77872, 283047, 158045, 137545, and 218178 particles contributed to the final reconstructions, from 1606,  1801, 1168, 1793, and 827 micrographs, respectively. The overall size and dataset associated with high-resolution reconstructions have been widely acknowledged for characterizing structural determination, particle orientation distribution and exploring particle defocus ranges.<br>For cryo-EM single particle control data, 2821 apo-ferrtin micrographs from EMPIAR-10424 on suspending ice , 3275 apo-ferrtin micrographs, 2209 ACE2 micrographs from conventional sample preparation method on supporting films were analyzed and compared with the ESI-cryoPrep dataset. The sample sizes for datasets were determined using the entire dataset contributing to the ultimate high-resolution reconstruction.<br>For cryo-ET analysis, four datasets of 12, 8, 6, and 18 tomograms were analyzed for 70S ribosome, 20S proteasome, apo-ferritin, and ACE2, respectively. The overall sample size is commonly employed for characterizing particle spatial distribution. |
| Data exclusions | For cryo-EM reconstruction, particles classified into bad classes with poorly defined features were excluded. To calculate the ice thickness of sample holes, we set a criteria to measure the z-height range of 85% particles to exclude outliers. |
| Replication | To demonstrate the reproducibility of the method and its benefits, we tested five sample proteins encompassing a large range of molecular sizes. All trials and attempts went successful. |
| Randomization | Representative tilt-series and negative staining micrographs were taken in a randomized way, we stochastically sampled the areas to record. During 3D refinement in 3D reconstruction softwares, particles were randomly divided into two subsets and independently reconstructed to determine the final resolution. |

| Blinding | Blinding experiments were not applicable to the current study since the treatments between the control data and the experiment data are the same. |
|----------|---|

# Reporting for specific materials, systems and methods

We require information from authors about some types of materials, experimental systems and methods used in many studies. Here, indicate whether each material, system or method listed is relevant to your study. If you are not sure if a list item applies to your research, read the appropriate section before selecting a response.

## Materials & experimental systems

| n/a | Involved in the study |
|-----|----------------------|
| ☒ ☐ | Antibodies |
| ☒ ☐ | Eukaryotic cell lines |
| ☒ ☐ | Palaeontology and archaeology |
| ☒ ☐ | Animals and other organisms |
| ☒ ☐ | Clinical data |
| ☒ ☐ | Dual use research of concern |

## Methods

| n/a | Involved in the study |
|-----|----------------------|
| ☒ ☐ | ChIP-seq |
| ☒ ☐ | Flow cytometry |
| ☒ ☐ | MRI-based neuroimaging |

