## [Peer Review File · Nature Methods]

Peer Review Information

Manuscript Title: Electropray-Assisted Cryo-EM Sample Preparation to Mitigate Interfacial Effects

Corresponding author name(s): Xiaoyu Zhou, Zheng Ouyang, Hong-Wei Wang

Editorial Notes: None

Reviewer Comments & Decisions:

Decision Letter, initial version:

Dear Hong-Wei,

Your Article, "Electropray-Assisted Cryo-EM Sample Preparation to Mitigate Interfacial Effects", has now been seen by 3 reviewers. As you will see from their comments below, although the reviewers find your work of considerable potential interest, they have raised a number of concerns. We are interested in the possibility of publishing your paper in Nature Methods, but would like to consider your response to these concerns before we reach a final decision on publication.

We therefore invite you to revise your manuscript to address these concerns. We are happy to see the overall positive response but we will strongly encourage adding another challenging example, if you already have such data or if it wouldn't take too long to collect, in addition to addressing all the other reviewer comments.

- * include a point-by-point response to the reviewers and to any editorial suggestions
- * please underline/highlight any additions to the text or areas with other significant changes to facilitate review of the revised manuscript
- * address the points listed described below to conform to our open science requirements
- * ensure it complies with our general format requirements as set out in our guide to authors at

www.nature.com/naturemethods

* resubmit all the necessary files electronically by using the link below to access your home page

[Redacted]

We hope to receive your revised paper within 8 weeks. If you cannot send it within this time, please let us know. In this event, we will still be happy to reconsider your paper at a later date so long as nothing similar has been accepted for publication at Nature Methods or published elsewhere.

OPEN SCIENCE REQUIREMENTS

REPORTING SUMMARY AND EDITORIAL POLICY CHECKLISTS

IMAGE INTEGRITY

When submitting the revised version of your manuscript, please pay close attention to our Digital Image Integrity Guidelines and to the following points below:

- that unprocessed scans are clearly labelled and match the gels and western blots presented in figures.
- that control panels for gels and western blots are appropriately described as loading on sample processing controls

-- all images in the paper are checked for duplication of panels and for splicing of gel lanes.

DATA AVAILABILITY

We strongly encourage you to deposit all new data associated with the paper in a persistent repository where they can be freely and enduringly accessed. We recommend submitting the data to discipline-specific and community-recognized repositories; a list of repositories is provided here:

<http://www.nature.com/sdata/policies/repositories>

All novel DNA and RNA sequencing data, protein sequences, genetic polymorphisms, linked genotype and phenotype data, gene expression data, macromolecular structures, and proteomics data must be deposited in a publicly accessible database, and accession codes and associated hyperlinks must be provided in the "Data Availability" section.

Please include a "Data availability" subsection in the Online Methods. This section should inform readers about the availability of the data used to support the conclusions of your study, including accession codes to public repositories, references to source data that may be published alongside the paper, unique identifiers such as URLs to data repository entries, or data set DOIs, and any other statement about data availability. At a minimum, you should include the following statement: "The data that support the findings of this study are available from the corresponding author upon request", describing which data is available upon request and mentioning any restrictions on availability. If DOIs are provided, please include these in the Reference list (authors, title, publisher (repository name), identifier, year). For more guidance on how to write this section please see:

<http://www.nature.com/authors/policies/data/data-availability-statements-data-citations.pdf>

CODE AVAILABILITY

Please include a "Code Availability" subsection in the Online Methods which details how your custom code is made available. Only in rare cases (where code is not central to the main conclusions of the paper) is the statement "available upon request" allowed (and reasons should be specified).

For more information on our code sharing policy and requirements, please see:
<https://www.nature.com/nature-research/editorial-policies/reporting-standards#availability-of-computer-code>

MATERIALS AVAILABILITY

SUPPLEMENTARY PROTOCOL

To help facilitate reproducibility and uptake of your method, we ask you to prepare a step-by-step Supplementary Protocol for the method described in this paper. We encourage authors to share their step-by-step experimental protocols on a protocol sharing platform of their choice and report the protocol DOI in the reference list. Nature Portfolio 's Protocol Exchange is a free-to-use and open resource for protocols; protocols deposited in Protocol Exchange are citable and can be linked from the published article. More details can found at www.nature.com/protocolexchange/about.

ORCID

Nature Methods is committed to improving transparency in authorship. As part of our efforts in this direction, we are now requesting that all authors identified as 'corresponding author' on published papers create and link their Open Researcher and Contributor Identifier (ORCID) with their account on the Manuscript Tracking System (MTS), prior to acceptance. This applies to primary research papers only. ORCID helps the scientific community achieve unambiguous attribution of all scholarly contributions. You can create and link your ORCID from the home page of the MTS by clicking on 'Modify my Springer Nature account'. For more information please visit please visit www.springernature.com/orcid.

Sincerely,
Arunima

Arunima Singh, Ph.D.
Senior Editor
Nature Methods

Reviewers' Comments:

Reviewer #1:

Remarks to the Author:

This manuscript by Yang et al. described a new cryo-EM grids preparation technique, by integrating the ESI device that used in native mass spectrometry to generate spray of nano-droplets for depositing protein sample in solution onto EM grids for plunge freezing. This method eliminates the need of blotting grids with filter paper, which has been a major factor influence the quality of protein samples being frozen. All cryo-EM reconstructions presented in this manuscript demonstrate that this sample preparation approach is suitable for robust cryo-EM grids preparation.

This is very interesting and exciting new development of preparing cryo-EM grids. Comparison with previous spray approach, such as the one developed by Joachim Frank lab for time resolved cryo-EM and the one developed by Bridget Carragher lab, the utilization of nano-ESI generate much smaller droplets, thus making the direct plunge freezing more reliable for generating vitreous ice of suitable ice thickness. Overall, the manuscript is well written, and the method presented is also exciting. I only have a few very minor specific comments, that I believe can be easily addressed by revision:

The theoretic description of droplet formation and charge distribution is interesting to me, but I am not sure how much this is to more general audience. I understood that droplet spreading on grids is a major issue for spray methods. In the case of Spotiton method, one has to use nano-wired grids to facilitate spreading. In this case, the spreading is facilitated by charges of individual droplet (if I understand this correctly). Thus, the theoretic description is interesting. My guess the reason it is in the Discussion rather than the main body of the manuscript is that such discussions are really descriptive rather than something that can be validated experimentally. I would still suggest moving this part to main body of the manuscript and explain the difference between this ESI spray and the regular spray.

Authors described optimizing the conditions for several test specimens. Are final datasets from each individual sample collected from grids prepared under the same/similar condition (except perhaps the protein concentration)? The question related is if the conditions identified/presented here are rather general to all different samples (as far as authors' own experience) or user would need to optimize these conditions for each individual sample?

My own experience of working with native mass spec is that ammonia buffer required here is sometimes or often harmful for protein complexes and one needs to optimize this buffer. Authors may also want to comment on that.

Some descriptions around formula (1) are unnecessarily confusing. What do authors mean by "manipulating the focal length"? I thought CTF refinement is just to refine the defocus value for individual particles, and the differences of defocus between different particles in the same

micrograph/movie give the difference in their respective z heights. Also, what do you mean by "measure ice thickness in the completed SPA dataset"?

Yifan Cheng

Reviewer #2:

Remarks to the Author:

This manuscript describes a fascinating new approach to the preparation of cryoEM samples. Sample prep remains a critical issue in the structure-determination pipeline, and new approaches are definitely needed.

The problem is ensuring that the protein sample is suspended between the air-water and substrate-water interfaces, to avoid preferential orientation of protein particles, and to avoid denaturation at the surfaces. The authors show a spectacular confinement of particles to intermediate planes in Fig. 4. (However Extended Fig. 4 shows ribosomes predominantly at the graphene-water interface. This is a very puzzling discrepancy.)

The best existing method involves spraying and freezing the sample on a time scale <100 ms, apparently too short for much diffusional rearrangement of macromolecules. The current method seems to do even better. The explanation given is that the macromolecules are repelled at the charged interfacial layers and remain in the center of the ice layer (lines 353-359). Could the authors give a reference for this theory, or more of a physical explanation?

As to the methods aspect, it is fascinating that the electrospray technology can be applied to this purpose. An important difference between cryoEM preparation and mass spec is that the former requires the macromolecules to remain hydrated in small droplets of buffer. Could you explain more the modifications you made to the MS application, and the reasoning behind them? All I can see is that you eliminated the N₂ gas flow (line 137).

I have two major concerns.

1. How quickly are the grids frozen after being sprayed? Is it within 100 ms, like the Spotiton? It is not clear how the grid is transported into the liquid ethane in drawing of Fig. 1a. As the explanation is given that it is electric-field effects that confine the particles to the middle of the solution layer, do you maintain the electric field up to the instant of freezing, to avoid diffusional disturbance of the layers described in Fig. 6?
2. The confinement of particles is said to arise from electric double layers set up by the applied electric field on the metal-gas-metal capacitor (lines 337-352). But a field of 3 kV/cm is only 0.3 mV/nm; this plus the high dielectric constant of water would seem to mean that field-induced electrostatic effects in the ~ 10 nm water film on the grid will be orders of magnitude smaller than kT/e , even in the Debye-Huckel region.

Other comments

Line 142, nano-ESI. You give the dimension of the capillary tip, but what is the size of the conventional ESI tip? And, in Fig. 1a, why is there such a long spray needle? Is the electrospray formed at the "spray ion source" or at the tip? Also, you compare the choice of metal coating, but I want to know, what advantages or drawbacks does nano-ESI give? Which do you recommend, ESI or nano-ESI for future work? What about flow rates? Are the effects of other variables about the same as for ESI?

Line 172, obtained resolutions. It would be helpful to give a rough idea of the dataset sizes (number of

particles or number of micrographs) at this point.

Line 184-185. "...compared to conventionally-prepared..." but you provide no comparison just the values from your specimens.

Line 221, eqn. (1). This equation, and the reference given, are more likely to cause confusion than illumination to cryoEM practitioners.

Line 274, "The present invention..." This is strange text for a research paper, but is appropriate for a patent application. The following text (lines 279-359) seems also to be like the very formal text of a patent application. If this work is related to a patent application, it would be appropriate to reference it.

Fig. 5d. Why is the ribosome angle distribution still not very good? And, if you did get a high-quality reconstruction from this dataset, please explain why. Were there sufficient images in between the dark areas to fill in the angleRot distribution?

Minor comments

Line 411. 8k x 8k K3 camera? Maybe give the correct physical pixel number and point out that you ran it in super-res mode.

Line 452. Definition of Euler angles here is unnecessary, as heat maps like these are well known in the cryoEM literature.

Fig. 1a. The inset picture is very important and too small! It's hard to see the scale bar, and if those are holey-carbon or holey gold holes, the scale bar seems to be wrong. And it's difficult to tell if there are smaller droplets. Somewhere please give a size range of the observed droplets. And, what do you see through the laser and camera setup?

Fig. 1c. This panel is really not informative. But, instead, why not give a closeup of the tip, grid, laser etc. which would be much more helpful.

Fig. 2. At first I didn't see the scale bars at all.

Fig. 5, Z-height distribution. Please define the polarity of these values; that is, what does a positive Z-height refer to: toward the air-water interface or toward the graphene?

Ext. Fig. 1. Please make it easier to notice that the bottom row is for the nano-ESI method.

Ext Fig. 2.- please label the abscissa in the plots; as it is, the titles, e.g. "Sample flow rate" seem at first to be the quantity plotted.

- Are the double-peaks in the distributions undesirable? Do they represent different charged species?

- What do the M.W. numbers refer to? Are they somehow computed from the top panel of each set of graphs?

Ext fig. 3a. Why is the ribosome result so different from Fig. 4?

Ext fig. 3b. Please increase the contrast (perhaps with lowpass filtering) as nothing is visible.

Ext fig. 5. Streptavidin?? Please mention in the text and methods what you did with this prep.

"adsorbed", "absorbed". You seem to use these two very different words interchangeably. In every case I noticed, "adsorbed" would be the correct word.

Signed,
Fred Sigworth

Reviewer #3:

Remarks to the Author:

Yang and Fan et al describe a highly novel method incorporating techniques from native mass spectrometry for cryo-EM grid preparation. The manuscript is well written and exhaustive in its

characterization of the technique and provides a new avenue for dealing with challenging samples. Although the authors have not described the utility of this method in samples that are truly difficult to work with using current conventional methodologies for cryo-EM grid preparation, the use of 4 well-studied specimens provides suitable guidelines for adapting these methods to other specimens. However, the authors still need to address a few technical details before it is suitable for publication. These mostly pertain to the claim that the method is suitable for overcoming the air-water interface problem. Although sufficient evidence has been presented in this work to support the claim, the authors need to validate the quality of the maps with additional metrics as listed below.

Major comments-

1. All of the described maps in this study need validation using the 3D-FSC tool to estimate quality of directional resolution with and without the use of ESICryo-prep methodology. Alternatively, the authors should at least provide map-model FSCs since atomic models are available for all 4 specimens. This would serve as an important metric to gauge whether the preferred orientation issue is indeed mitigated at the level of map quality. Although apoferritin and proteasome are not expected to see significant differences in these metrics, the ACE2 and ribosome samples should in theory have much better metrics as partially evidenced by figure 5d.

Minor comments-

1. Line 26, change 'finely tuned' to 'fine-tuned'.
2. Line 52, 83-84, describe an actual suitable temperature range, since at true liquid nitrogen temperature, the widely used cryogen liquid ethane tends to solidify.
3. Line 55, 69, change 'structural' to 'structure'.
4. Line 85, change 'blotting free' to 'blot-free'.
5. Lines 125-126, it appears there seems to be an optimum with respect to flow rate for apoferritin. Why do the authors conclude that decreasing flow rate impacts particle quality? Only 100 nL/min appears to be optimal with flow rates on either side of the the optimum flow rate showing less intact particles. Authors should amend this statement accordingly.
6. Extended figure 3 would be more complete if compared with proportion of particles used for reconstructions for grids prepared using standard techniques.
7. Extended figure 4a, why is the apoferritin Z-X slice smeared compared to the other images? The tomogram has an inherent tilt and appears stretched. Please verify if the tilt alignment scheme was optimal for this experiment.
8. Line 194, figure 4d is mislabeled, should be 5d.
9. Line 205, figure 3a is mislabeled, should be 4a.
10. Figure 5, can authors expand on the differences in the apparent skewness values in panel a versus panel b. How do the values scale? For example, for apoferritin without supporting film, negative skewness is -1.92 and positive skewness is 2.63 in panel a, yet in panel b the positive skewness appears to be larger than 5 and negative skewness closer to -10.
11. Line 385-386, authors should expand on the buffer conditions of the 4 specimens. It is implied that ammonium acetate alone is the dominant solvent, but it is unclear how much of the other buffer components that were present during purification are still present in solution. These details can help readers by serving as guidelines for optimal sample preparation. In addition, please provide exact concentrations of the specimens used for each experiment.
12. Table 1, authors should clarify the nominal magnification for ribosome, proteasome and apoferritin. Relative to the pixel size and magnification of imaging conditions for ACE2, the numbers don't seem accurate.
13. The authors should emphasize the molecular weight of the each of the 4 samples, as it is seemingly impressive to have this method work for a wide range of sizes

Author Rebuttal to Initial comments

We appreciate the comprehensive and positive feedback of the editor and reviewers on our ESI-cryoPrep manuscript. The three reviewers posed insightful questions, contributing to the manuscript's overall enhancement, and aligning with our objectives. We have read carefully and revised our manuscript accordingly.

Please find our detailed responses to each point of the editor and reviewers below. The editor's and reviewers' comments are in *italic font*.

Reviewer #1:

Remarks to the Author:

This manuscript by Yang et al. described a new cryo-EM grids preparation technique, by integrating the ESI device that used in native mass spectrometry to generate spray of nano-droplets for depositing protein sample in solution onto EM grids for plunge freezing. This method eliminates the need of blotting grids with filter paper, which has been a major factor influence the quality of protein samples being frozen. All cryo-EM reconstructions presented in this manuscript demonstrate that this sample preparation approach is suitable for robust cryo-EM grids preparation.

This is very interesting and exciting new development of preparing cryo-EM grids. Comparison with previous spray approach, such as the one developed by Joachim Frank lab for time resolved cryo-EM and the one developed by Bridget Carragher lab, the utilization of nano-ESI generate much smaller droplets, thus making the direct plunge freezing more reliable for generating vitreous ice of suitable ice thickness. Overall, the manuscript is well written, and the method presented is also exciting. I only have a few very minor specific comments, that I believe can be easily addressed by revision:

The theoretic description of droplet formation and charge distribution is interesting to me, but I am not sure how much this is to more general audience. I understood that droplet spreading on grids is a major issue for spray methods. In the case of Spotiton method, one has to use nano-wired grids to facilitate spreading. In this case, the spreading is facilitated by charges of individual droplet (if I understand this correctly). Thus, the theoretic description is interesting. My guess the reason it is in the Discussion rather than the main body of the manuscript is that such discussions are really descriptive rather than something that can be validated experimentally. I would still suggest moving this part to main body of the manuscript and explain the difference between this ESI spray and the regular spray.

Response:

We concur with the Reviewer's observation regarding the placement of the theoretical description of droplet formation and charge distribution. We have updated our manuscript by introducing a new mechanism, the charge residue model, in place of the electric double layer model. This modification is well-established and commonly employed in the field of mass spectrometry to elucidate similar phenomena. The goal of this revision is to enhance the clarity and comprehension of our hypothesis. We now have moved the theoretical description in the Results, as shown below:

The present cryo-sample preparation technique utilizing ESI involves the generation of charged microdroplets from a protein solution¹. ESI operates at atmospheric pressure where the analyte solution is infused into a metal capillary held at a positive electric potential of several thousand volts. The solution forms a Taylor cone at the capillary tip, emitting fine mist droplets when the Raleigh limit is reached. These initial droplets, with micrometer-scale radii, carry an excess of positive charge primarily from protonated ions^{2,4}. The charge density significantly increases during solvent evaporation as they move towards the supporting substrate (Fig. 6a). It is established in the ESI process that small ions are inclined to accumulate at the surface of charged droplets, eventually transforming into dry ions through ion evaporation^{2,5}. In contrast, large, multiply-charged, and highly hydrophilic protein ions reside at the center, becoming dry ions after the evaporation of water and small ions. This phenomenon is known as the charge residue model (CRM)⁶⁻⁸. In the ESI-cryoPrep method, the CRM mechanism aids in distributing protein ions away from both the air-water and graphene-water interfaces upon landing on the supporting substrate, ultimately preserving their native forms.

Revised Fig. 6: Mechanism of spreading and wetting of spray droplets, and the charge residue model at interfaces.

(a) Schematic representation of the evolution of electrified droplets in the ESI-cryoPrep equipment.

(b) The structure of the ESI-cryoPrep device could be considered an electrochemical cell, with negative charges accumulating on the surface of the conductive supporting grid.

(c) Adjacent droplets form a water bridge and merge into a larger droplet through wetting effects. As the droplets merge, their surface area decreases, which leads to a decrease in the total surface energy. The remaining energy is liberated.

(d) The schematic representation of the charge residue model at the air-water or graphene-water interfaces serves as a barrier, preventing direct interactions between protein particles and the opposing phase. Polar water solvents demonstrate robust interactions with the charged droplet

surface or the charged electrode. The radial charge gradient arises from mutual coulombic repulsion among excess charges within the charge residue model of droplets. The upper panel includes legends for various ions.

As shown in the main text, the ESI-cryoPrep approach proficiently confines macromolecules within the core of vitreous ice, preventing the adherence of target molecules to the air-water or graphene-water interfaces. This phenomenon is presumably attributed to uneven ion distributions in terms of particle sizes and charges^{3,9-11}, akin to mechanisms discussed earlier for ion evaporation and charge residual models in the ESI process. Merged droplets on the substrate retain excessive charges, preventing protein ions from reaching air-water and substrate-water interfaces, as depicted in Fig. 6d.

We have also compiled a summary of the distinct Electrospray Ionization (ESI) parameters used in Mass Spectrometry (MS) analysis and our cryo-EM specimen preparation, as presented in the table below.

Supplementary Table 3. Comparisons of ESI conditions in MS analysis and cryo-EM sample preparation

ESI components	MS analysis	Cryo-EM sample preparation
Capillary voltage	4000-5000 V	4000-5000 V
Nebulizer	0.7 mBar	0 mBar
Desolvation dry gas	0-6 L/min	0 L/min
Desolvation temperature	20-50 °C	Room temperature
Sample flow rate	0.5-5 µl/min	0.3-0.5 µl/min

In deviation from conventional Mass Spectrometry conditions, modifications were implemented in ESI for cryo-EM sample preparation, employing a softer approach for proteins and protein complexes. In the revised manuscript, Supplementary Table 3 outlines adjustments made to temperature, gas flow, and ionization voltage. N₂ dry gas flow was eliminated to prevent protein unfolding and supercharging, and reduced ionization voltages were employed to mitigate electrochemical effects impacting protein structure and intensity distribution.

Authors described optimizing the conditions for several test specimens. Are final datasets from each individual sample collected from grids prepared under the same/similar condition (except perhaps the protein concentration)? The question related is if the conditions identified/presented here are rather general to all different samples (as far as authors' own experience) or user would need to optimize these conditions for each individual sample?

Response:

We appreciate the Reviewer for raising this concern, as it is important for the generalization of the method. Through systematic exploration and adjustment of various parameters and sampling

conditions in electrospray ionization (ESI), we identified key variables impacting the preservation of intact protein structures. We believe that these factors are generalized for most protein cases. While most preparation conditions are uniform across various macromolecule samples, our current emphasis is on adjusting protein concentrations.

My own experience of working with native mass spec is that ammonia buffer required here is sometimes or often harmful for protein complexes and one needs to optimize this buffer. Authors may also want to comment on that.

Response:

We thank the Reviewer for highlighting this concern, as these details can serve as valuable guidelines for readers seeking optimal sample preparation. In the initial stages of our experiments, negative staining EM (Fig. R1-1) demonstrated that conventional protein buffers containing salts, such as NaCl and PBS, resulted in salt crystal formation in the sample due to the desolvation effect of charged droplets in flight. To address electrolyte concentration and prevent salt crystallization caused by desolvation, we substituted NaCl with volatile ammonium acetate salt in the buffer solution, successfully eliminating salt crystals in the samples. The five tested proteins maintained their integrity and structure without significant alterations when using the ammonium acetate-Tris buffer. We are in the process of further optimizing the protocols for more buffer conditions in our setup.

Fig. R1-1: Salt crystal formation in certain ESI samples

(a) Concentrated salt crystals observed through negative staining EM following the ejection of a 150 mM NaCl sample at the electrolyte concentration of a typical protein buffer. (b) Concentrated salt crystals resulting from a 130 mM PBS sample were observed through negative staining EM at a relatively low magnification. Crystallizable salts precipitate into salt crystals at elevated concentrations, posing challenges to the preservation of proteins in their physiological state.

Some descriptions around formula (1) are unnecessarily confusing. What do authors mean by “manipulating the focal length”? I thought CTF refinement is just to refine the defocus value for individual particles, and the differences of defocus between different particles in the same micrograph/movie give the difference in their respective z heights.

Response:

We appreciate the Reviewer for highlighting this matter, which stems from a misphrasing in our previous version. We have removed the equation to eliminate ambiguity in the revised manuscript. The expression "manipulating the focal length" refers to the calibration of the z-axis height of the grid. Typically, adjustments are made by implementing a stage shift when is substantial or by modifying the current of the C2 lens when defocus is minimal. Thus, variations in particle defocus align with their respective Z-heights in vitreous ice. This aligns with the reviewer's description and we have revised the content accordingly.

Also, what do you mean by "measure ice thickness in the completed SPA dataset"?

Response:

Exclusively relying on tomography to gauge ice thickness may introduce bias and uncertainty due to inherent flatness variation across the grid. To address this, a more robust and statistically sound approach involves leveraging a larger dataset. Because the ice thickness revealed by tomography is less statistical, we established a comprehensive ice thickness analysis within the complete single particle analysis dataset by evaluating relative Z-heights derived from defocus variations. While considering particle diameter would be more precise, this measurement provides a generalized depiction of sample ice thickness.

Yifan Cheng

Reviewer #2:

Remarks to the Author:

This manuscript describes a fascinating new approach to the preparation of cryoEM samples. Sample prep remains a critical issue in the structure-determination pipeline, and new approaches are definitely needed.

The problem is ensuring that the protein sample is suspended between the air-water and substrate-water interfaces, to avoid preferential orientation of protein particles, and to avoid denaturation at the surfaces. The authors show a spectacular confinement of particles to intermediate planes in Fig. 4. (However Extended Fig. 4 shows ribosomes predominantly at the graphene-water interface. This is a very puzzling discrepancy.)

Response:

We appreciate the Reviewer's comments. We thank the reviewer for indicating the Extended Data Fig. 4 problem, which stemmed from a misoperation during the tomographic reconstruction presentation. In the case of the tomogram featured in Extended Figure 4a, the reconstruction of a 70S ribosome tomogram was executed using IMOD. However, in the "Get XYZ Volume Range" step of 3dmod, where the Z-axis range is established to exclude non-sample material, an unintentionally narrow Z-axis range was set. This oversight led to the exclusion of certain slices from the tomogram. Figure 4 and Extended Figure 4a both consistently reveal the same findings. Here, in Fig. R2-1 (Revised Extended Data Fig. 5a), we introduce an additional 70S ribosome tomogram, illustrating particle distribution both away from the air-water interface and the graphene-water interfaces.

Fig. R2-1: Tomogram of 70S ribosome in the ESI-cryoPrep specimen.

The best existing method involves spraying and freezing the sample on a time scale <100 ms, apparently too short for much diffusional rearrangement of macromolecules. The current method seems to do even better. The explanation given is that the

macromolecules are repelled at the charged interfacial layers and remain in the center of the ice layer (lines 353-359). Could the authors give a reference for this theory, or more of a physical explanation?

Response:

After thorough consideration, we have opted to employ a well-known mechanism to elucidate the distribution of the proteins inside the droplets. This mechanism has been used to support the charge residue model in the ESI process. Electrospray produces excessively charged droplets, inside which the small ions are inclined to accumulate at the surfaces of the droplets, while large protein ions reside at the center. Subsequently, small ions evolve into dry ions through ion evaporation, while large protein ions transition into dry ions following the evaporation of water molecules and small ions (as per the charge residue model) in ESI of mass spectrometry. The new model and references have been provided in our response to reviewer 1.

As to the methods aspect, it is fascinating that the electrospray technology can be applied to this purpose. An important difference between cryoEM preparation and mass spec is that the former requires the macromolecules to remain hydrated in small droplets of buffer. Could you explain more the modifications you made to the MS application, and the reasoning behind them? All I can see is that you eliminated the N₂ gas flow (line 137).

Response:

The revised manuscript now provides clarifications regarding the modifications made to conventional ESI for Cryo-EM sample preparation, highlighting the differences compared to Mass Spectrometry (MS) applications, as presented in Supplementary Table 3 in the revised manuscript. Please also see our response to reviewer 1 above.

I have two major concerns.

1. How quickly are the grids frozen after being sprayed? Is it within 100 ms, like the Spotiton? It is not clear how the grid is transported into the liquid ethane in drawing of Fig. 1a. As the explanation is given that it is electric-field effects that confine the particles to the middle of the solution layer, do you maintain the electric field up to the instant of freezing, to avoid diffusional disturbance of the layers described in Fig. 6?

Response:

We have revised Fig. 1a (Fig. R2-2 below) to include an illustration of the manual operation procedure at the bottom left. The grids were manually transferred to liquid ethane, with a recorded time of approximately 490 ms ± 70 ms (Fig. R2-3 below) from transfer to freezing. Each experiment consisted of 50 replicates.

Fig. R2-2 Schematic of the ESI-cryoPrep device.

Fig. R2-3 Statistical analysis of spot-to-plunge time with 50 replicates.

As for the electric field built-up, the ESI-cryoPrep apparatus can be conceptualized as a metal-gas-metal capacitor, resembling an energy storage battery, where the application of voltage charges the EM mesh connected to negative electrodes. Rapid equilibrium in capacitor charging is facilitated by the electric conductivity of the carrier grid, graphene, and metal tweezers, resulting in a surface accumulation of negative charge on the carrier grid. In the manual transfer operation of the carrier grid, taking approximately 500 ms with insulated gloves, we anticipate no charge transfer or neutralization.

In recent attempts at cryo-sample preparation, the negative electrode was directly attached to metal tweezers to prevent charge spreading from leakage currents during EM grid transfer. The negative electrode remained connected to the carrier grid throughout the transfer process. The

negative charge on the EM grid (electric field) was kept until the sample freezing and was not harmful to the human operator.

2. The confinement of particles is said to arise from electric double layers set up by the applied electric field on the metal-gas-metal capacitor (lines 337-352). But a field of 3 kV/cm is only 0.3 mV/nm; this plus the high dielectric constant of water would seem to mean that field-induced electrostatic effects in the ~10 nm water film on the grid will be orders of magnitude smaller than kT/e , even in the Debye-Huckel region.

Response:

We thank the reviewer for pointing this out. This question has been addressed in our previous responses to both reviewer 1 and this reviewer.

Other comments

Line 142, nano-ESI. You give the dimension of the capillary tip, but what is the size of the conventional ESI tip? And, in Fig. 1a, why is there such a long spray needle? Is the electrospray formed at the "spray ion source" or at the tip?

Response:

1. The conventional ESI tip has an outer diameter of approximately 200 μm .
2. The length of the spray needle influences the efficiency of the electrospray process. Longer needles offer enhanced control of the electric field, leading to the production of finer, more uniform droplets. This improvement contributes to the overall performance of the electrospray system, enhancing ionization efficiency and stability.
3. Electrospray initiation occurs at the tip of the ESI needle.

Also, you compare the choice of metal coating, but I want to know, what advantages or drawbacks does nano-ESI give? Which do you recommend, ESI or nano-ESI for future work? What about flow rates? Are the effects of other variables about the same as for ESI?

Response:

We appreciate the reviewer for raising this concern.

With an outer diameter in the order of tens of nanometers, the nanospray needle generates nanoscale initial droplets with high desolvation tendency, facilitating the production of extremely small final droplets for the analysis of ultrathin samples.

Nanospraying requires low sample concentrations (μM) and minimal volume. In nano-ESI, the protein solution is injected into the nano-ESI needle prior to the application of voltage and the spraying process, resulting in constrained control over the flow rate compared to ESI (low flow rate 20–50 nL/min). Typically, a spray voltage of 0.7–1.1 kV is applied through an electrically conducting coating, often a sputtered gold film, on the capillary's outer surface. Upon activating the high voltage, the analyte solution flow is driven solely by capillary forces refilling the aperture as droplets exit the tip. In contrast to conventional ESI, which generates initial charged droplets of 1–2 μm diameter, nano-ESI produces charged droplets less than 200 nm in diameter—approximately 100–1000 times smaller in volume than those from conventional micro-emitters.

Yet, NanoESI is currently challenged by significant electrochemical effects, causing particle dissociation and potential vaporization of the thin fluid film under current preparation conditions.

ESI and nano-ESI are suitable for distinct working contexts. nano-ESI shows potential under softer electrospraying conditions, which are yet to be determined. For the current analysis of protein structures, opting for conventional ESI would be more advisable.

Line 172, obtained resolutions. It would be helpful to give a rough idea of the dataset sizes (number of particles or number of micrographs) at this point.

Response:

As requested, we have provided the dataset sizes in the Supplementary Table 1 of the revised manuscript.

	70S ribosome	20S proteasome	Apo-ferritin	ACE2	Streptavidin
No. of micrographs	1272	1743	946	1557	840
No. of particles (contributed to final map)	74,004	269,435	148,352	126,622	212,011

Line 184-185. "...compared to conventionally-prepared..." but you provide no comparison just the values from your specimens.

Response:

We now have updated Extended Data Figure 3 in the revision for the comparison.

Revised Extended Data Fig. 3 Proportion of particles contributing to high-resolution reconstruction.

The proportion of particles contributed to high-resolution reconstruction. Particles of 70S ribosome and ACE2 using the conventional method on supporting films experienced significant preferential orientation issues, leading to the failure to reconstruct satisfactory 3D maps.

Line 221, eqn. (1). This equation, and the reference given, are more likely to cause confusion than illumination to cryoEM practitioners.

Response:

This was a misphrasing. We have removed the equation to eliminate ambiguity in the manuscript. Typically, adjustments are made by implementing a stage shift when the defocus is substantial or by modifying the current of the C2 lens when the defocus is minimal. Thus, variations in particle defocus from CTF refinement align with their respective Z-heights in vitreous ice.

Line 274, "The present invention..." This is strange text for a research paper, but is appropriate for a patent application. The following text (lines 279-359) seems also to be like the very formal text of a patent application. If this work is related to a patent application, it would be appropriate to reference it.

Response:

We have revised the text to be more aligned with the research article format.

Fig. 5d. Why is the ribosome angle distribution still not very good? And, if you did get a high-quality reconstruction from this dataset, please explain why. Were there sufficient images in between the dark areas to fill in the angleRot distribution?

Response:

Thank the reviewer for pointing this out. We have a hypothesis to explain this phenomenon. In an aqueous solution devoid of interfacial effects, protein particles undergo Brownian motion, engaging in perpetual irregular motion due to collisions with liquid molecules in all directions. Consequently, these particles exhibit diverse angular orientations. However, in actual cryo-samples, the protein complexes of varying sizes may not exhibit fully randomized movements as small ions. When confined in relatively thin ice, even if they are not influenced by the interfacial effects, the majority of large particles may adopt specific orientations. Nonetheless, successful 3D reconstructions can still be achieved for ribosomes because there is still quite a wide angular distribution of particles to fill in the 3D Fourier space.

Minor comments

Line 411. 8k x 8k K3 camera? Maybe give the correct physical pixel number and point out that you ran it in super-res mode.

Response:

We thank the Reviewer for pointing out this incorrect description in the original version.

The K3 direct detection camera captures images in super-resolution with a size of 11,520 × 8,184 pixels." We've rephrased the expression in the manuscript.

Line 452. Definition of Euler angles here is unnecessary, as heat maps like these are well known in the cryoEM literature.

Response:

We already removed the description of Euler angles and heat maps as suggested.

Fig. 1a. The inset picture is very important and too small! It's hard to see the scale bar, and if those are holey-carbon or holey gold holes, the scale bar seems to be wrong. And it's difficult to tell if there are smaller droplets. Somewhere please give a size range of the observed droplets. And, what do you see through the laser and camera setup?
 Fig. 1c. This panel is really not informative. But, instead, why not give a closeup of the tip, grid, laser etc. which would be much more helpful.

Response:

We have made the following revisions and clarifications:

Part of revised Fig. 1: Schematic representation of ESI-cryoPrep design and device.

(a) Arrangements of ESI-cryoPrep physical device. Device schematic, with an inset zooming in on the grid surface depicting deposited droplets. (c) Close-ups of the tip, grid, laser, and spray. Zoomed camera and laser inspection of grid surface showing deposition of sprayed droplets. (d) Sequential snapshots of the spray process were captured at 2, 4, 6, and 8 seconds. Scale bars are 500 μm.

The laser was used to monitor the Taylor cone of the spray, which was a critical criterion for ensuring the continuity of the spray. The camera was used to observe the spraying process and to determine the collection time of the spray, avoiding excessively thick ice layers that could affect sample observation as shown in Extended Data Fig. 7.

Extended Data Fig. 7 Camera inspection of grid surface showing deposition of sprayed droplets.

The spray processes were recorded at 2 sec, 4 sec, 6 sec, and 8 sec by CCD camera. a-d. Droplets were collected in four different grids.

Fig. 2. At first I didn't see the scale bars at all.

Response:

We thank the Reviewer for the suggestion and have corrected it in the revised version of the manuscript.

Fig. 5, Z-height distribution. Please define the polarity of these values; that is, what does a positive Z-height refer to: toward the air-water interface or toward the graphene?

Response:

As requested by the Reviewer, we have incorporated the polarities of Z-height distributions for the five ESI-cryoPrep specimens in the manuscript. The distribution with a higher proportion of skewed sides, signifying dominant adsorption, hints at potential adsorption at the air or graphene interface. However, the control data were sourced from EMPIAR or previous experiments in our lab, collecting only SPA datasets rather than tomography data. The absence of electron tomography results for these control data makes it challenging to precisely determine whether the positive and negative skews specifically relate to the air-water interface or the graphene-water interface.

Ext. Fig. 1. Please make it easier to notice that the bottom row is for the nano-ESI method.

Response:

We appreciate the Reviewer for the suggestion and have corrected it in the revised version of the manuscript.

Ext Fig. 2.- please label the abscissa in the plots; as it is, the titles, e.g. "Sample flow rate" seem at first to be the quantity plotted.

Response:

Extended Data Fig. 2 has been reproduced as suggested. The abscissa in Extended Figure 2 has been more clearly labeled, and all titles were initially labeled.

- Are the double-peaks in the distributions undesirable? Do they represent different charged species?

The observed double peaks are mainly attributed to intact proteins (m/z 8000~10000) and lost subunits (m/z 2000~4000) partially from denatured proteins. The presence of double peaks in the lost subunits part could potentially be improved by adjusting parameters such as sample flow rate and protein concentration.

- What do the M.W. numbers refer to? Are they somehow computed from the top panel of each set of graphs?

"M.W." is the abbreviation for molecular weight, calculated using the formula by $M.W. = m/z * z$, where m/z and z represent the mass-to-charge ratio and charge number of the proteins measured by mass spectrometry, respectively. The molecular weights presented here are derived from the mass spectrometry profiles.

Ext fig. 3a. Why is the ribosome result so different from Fig. 4?

Response:

This question has been addressed in our response to Question 1 of this reviewer.

Ext fig. 3b. Please increase the contrast (perhaps with lowpass filtering) as nothing is visible.

Response:

We appreciate the Reviewer for the suggestion and have corrected it in the revised version of the manuscript.

Ext fig. 5. Streptavidin?? Please mention in the text and methods what you did with this prep.

Response:

Initially, streptavidin was employed as a control dataset to illustrate particle adsorption at either the air-water or graphene-water interface. Even with the inclusion of a supporting film, researchers observed particle adsorption at the air-water interface, rendering them unsuitable for final high-resolution analysis. Responding to the editor's request for an additional case, we subsequently prepared an ESI-cryoPrep streptavidin sample and conducted the same analysis as with the other four samples. The corresponding results are appended to the manuscript.

"adsorbed", "absorbed". You seem to use these two very different words interchangeably. In every case I noticed, "adsorbed" would be the correct word.

Response:
Corrected.

*Signed,
Fred Sigworth*

Reviewer #3:**Remarks to the Author:**

Yang and Fan et al describe a highly novel method incorporating techniques from native mass spectrometry for cryo-EM grid preparation. The manuscript is well written and exhaustive in its characterization of the technique and provides a new avenue for dealing with challenging samples. Although the authors have not described the utility of this method in samples that are truly difficult to work with using current conventional methodologies for cryo-EM grid preparation, the use of 4 well-studied specimens provides suitable guidelines for adapting these methods to other specimens. However, the authors still need to address a few technical details before it is suitable for publication. These mostly pertain to the claim that the method is suitable for overcoming the air-water interface problem. Although sufficient evidence has been presented in this work to support the claim, the authors need to validate the quality of the maps with additional metrics as listed below.

Major comments-

1. All of the described maps in this study need validation using the 3D-FSC tool to estimate quality of directional resolution with and without the use of ESIcryo-prep methodology. Alternatively, the authors should at least provide map-model FSCs since atomic models are available for all 4 specimens. This would serve as an important metric to gauge whether the preferred orientation issue is indeed mitigated at the level of map quality. Although apoferritin and proteasome are not expected to see significant differences in these metrics, the ACE2 and ribosome samples should in theory have much better metrics as partially evidenced by figure 5d.

Response:

We thank the Reviewer for pointing out this comprehensive method for map validation and have included the results in the revised version of the manuscript. The 3D-FSC analysis of the four ESI-cryoPrep specimens and control data is presented below (Fig. R3-1, R3-2, R3-3, R3-4). As expected, owing to their high symmetry, the 20S proteasome and apo-ferritin exhibit slight improvements compared to the control data. In contrast, the 70S ribosome and ACE2 by ESI-cryoPrep demonstrate significant improvement of 3D-FSC in all Fourier directions than control. We revised the manuscript to include the 3DFSC analysis in Extended Data Fig. 4.

Fig. R3-1: The 3DFSC analysis of ESI-cryoPrep and control data for the 70S ribosome sample.

Fig. R3-2: The 3DFSC analysis of ESI-cryoPrep and control data for the 20S proteasome sample.

Fig. R3-3: The 3DFSC analysis of ESI-cryoPrep and control data for the apo-ferritin sample.

Fig. R3-4: The 3DFSC analysis of ESI-cryoPrep and control data for the ACE2 sample.

Minor comments-

1. Line 26, change 'finely tuned' to 'fine-tuned'.

Response:

Corrected.

2. Line 52, 83-84, describe an actual suitable temperature range, since at true liquid nitrogen temperature, the widely used cryogen liquid ethane tends to solidify.

Response:

We have included the suitable temperature range in our revised manuscript.

3. Line 55, 69, change 'structural' to 'structure'.

Response:

Corrected.

4. Line 85, change 'blotting free' to 'blot-free'.

Response:

Corrected.

5. Lines 125-126, it appears there seems to be an optimum with respect to flow rate for apoferritin. Why do the authors conclude that decreasing flow rate impacts particle quality? Only 100 nL/min appears to be optimal with flow rates on either side of the the optimum flow rate showing less intact particles. Authors should amend this statement accordingly.

Response:

We thank the Reviewer for pointing out this error and have corrected it in the revised version of the manuscript.

6. Extended figure 3 would be more complete if compared with proportion of particles used for reconstructions for grids prepared using standard techniques.

Response:

We thank the Reviewer for highlighting this problem and have updated our Extended Data Figure 3 as shown in our response to reviewer 2.

7. Extended figure 4a, why is the apoferritin Z-X slice smeared compared to the other images? The tomogram has an inherent tilt and appears stretched. Please verify if the tilt alignment scheme was optimal for this experiment.

Response:

We thank the Reviewer for pointing out this error and have corrected it in the revised version of the manuscript. We reproduced the tilt alignment and the results are the following.

Fig. R3-6: Apoferritin particle distributions in tomograms.

8. Line 194, figure 4d is mislabeled, should be 5d.

Response:
Corrected.

9. Line 205, figure 3a is mislabeled, should be 4a.

Response:
Corrected.

10. Figure 5, can authors expand on the differences in the apparent skewness values in panel a versus panel b. How do the values scale? For example, for apoferritin without supporting film, negative skewness is -1.92 and positive skewness is 2.63 in panel a, yet in panel b the positive skewness appears to be larger than 5 and negative skewness closer to -10.

Response:

The sample skewness is computed as the Fisher-Pearson coefficient of skewness, given by:

$$g_1 = \frac{m_3}{m_2^{3/2}}$$

where $m_i = \frac{1}{N} \sum_{n=1}^N (x[n] - \bar{x})^i$ is the biased sample i -th central moment, and \bar{x} is the sample mean. Alternatively, the calculations are corrected for bias, and the value computed is the adjusted Fisher-Pearson standardized moment coefficient:

$$G_1 = \frac{k_3}{k_2^{3/2}} = \frac{\sqrt{N(N-1)} m_3}{N-2 m_2^{3/2}}$$

As a general guideline, skewness values less than -1 or greater than 1 indicate a highly skewed distribution. Values falling between -1 and -0.5 or between 0.5 and 1 suggest moderate skewness, while those between -0.5 and 0.5 indicate an approximately symmetric distribution. A larger absolute skewness value signifies increased asymmetry in the distribution, with more prominent long tails on either side. This results in varying frequencies of gains or losses within each interval on both sides.

Fig. R3-7: Skewness distributions

In the case of apo-ferritin without a supporting film, most skewness values in the distribution fall within the range of -5 to +5. Outliers in the dataset contribute to a few extreme skewness values. As observed in the figure above, skewness values surpassing 0.5 or falling below -0.5 indicate noticeable asymmetry from both sides. Considering the limited occurrence of extreme skewness values, distributions with skewness within ± 5 provide a more accurate portrayal of the z-height distribution. The extreme outliers (cut-off=1%) have been excluded in the revised manuscript.

11. Line 385-386, authors should expand on the buffer conditions of the 4 specimens. It is implied that ammonium acetate alone is the dominant solvent, but it is unclear how much of the other buffer components that were present during purification are still present in solution. These details can help readers by serving as guidelines for optimal sample preparation. In addition, please provide exact concentrations of the specimens used for each experiment.

Response:

We appreciate the guidance provided by the reviewer and have addressed this in the revised version of the manuscript, as shown in Supplementary Table 2.

Supplementary Table 2. Cryo-sample preparation conditions

	70S ribosome	20S proteasome	apo-ferritin	ACE2	Streptavidin
Purified protein buffer conditions	20 mM HEPES-KOH, pH 7.6, 10 mM Mg(OAc) ₂ , 30 mM KCl, 7 mM beta-mercaptoethanol	50 mM Tris-HCl, pH 8.0, 100 mM NaCl	PBS buffer (50 mM NaH ₂ PO ₄ , pH 7.4, 150 mM NaCl)	HBS buffer (10 mM HEPES, pH 7.2, 150 mM NaCl)	140 mM NaCl, 8 mM Sodium Phosphate, 2 mM Potassium Phosphate, 10 mM KCl, pH 7.4
Dilution factor	30×	5×	10×	5×	8×
Dilution buffer	20 mM Tris, pH 8.0, 100 mM NH ₄ Ac				
Cryo-sample protein concentrations	0.44 μM	0.58 μM	1.2 μM	4 μM	2.5 μM
Molecular weight	2.5 MDa	750 kDa	444 kDa	70 kDa	52 kDa

12. Table 1, authors should clarify the nominal magnification for ribosome, proteasome and apoferritin. Relative to the pixel size and magnification of imaging conditions for ACE2, the numbers don't seem accurate.

Response:

It may be a bit confusing, as the data were obtained using different microscopes. The 70S ribosome, 20S proteasome, and apo-ferritin data were collected using a Titan Krios D3172 microscope (FEI) operating at 300 kV. For ACE2 data, a Titan Krios D3418 microscope (FEI) equipped with a GIF Quantum energy filter (slit width 20 eV) at 300 kV was employed. Due to the absence of an energy filter on Titan Krios D3172, the camera-to-column distance is shorter compared to Titan Krios D3418. Despite having the same current for both the intermediate and projection lenses, the nominal magnification on Titan Krios D3418 exceeds that of Titan Krios D3172.

As requested, we have provided the exact TEM imaging conditions in the revised manuscript.

13. The authors should emphasize the molecular weight of the each of the 4 samples, as it is seemingly impressive to have this method work for a wide range of sizes.

Response:

We have incorporated the suggested contents in the revised version of the manuscript (Supplementary Table 1).

References:

- 1 Kebarle, P. & Verkerk, U. H. Electrospray: from ions in solution to ions in the gas phase, what we know now. *Mass Spectrom Rev* **28**, 898-917 (2009).
<https://doi.org/10.1002/mas.20247>
- 2 Wilm, M. Principles of electrospray ionization. *Molecular & cellular proteomics* **10** (2011).
- 3 Kebarle, P. & Tang, L. From ions in solution to ions in the gas phase—the mechanism of electrospray mass spectrometry. *Anal Chem* **65**, 972A-986A (1993).
- 4 Konermann, L., Ahadi, E., Rodriguez, A. D. & Vahidi, S. Unraveling the mechanism of electrospray ionization. *Analytical chemistry* (2013).
- 5 Iribarne, J. & Thomson, B. On the evaporation of small ions from charged droplets. *The Journal of chemical physics* **64**, 2287-2294 (1976).
- 6 De La Mora, J. F. Electrospray ionization of large multiply charged species proceeds via Dole's charged residue mechanism. *Analytica Chimica Acta* **406**, 93-104 (2000).
- 7 Kebarle, P. & Verkerk, U. H. Electrospray: from ions in solution to ions in the gas phase, what we know now. *Mass spectrometry reviews* **28**, 898-917 (2009).
- 8 Iavarone, A. T. & Williams, E. R. Mechanism of charging and supercharging molecules in electrospray ionization. *Journal of the American Chemical Society* **125**, 2319-2327 (2003).
- 9 Kebarle, P. & Peschke, M. On the mechanisms by which the charged droplets produced by electrospray lead to gas phase ions. *Analytica Chimica Acta* **406**, 11-35 (2000).
- 10 Beveridge, R. *et al.* Ion mobility mass spectrometry uncovers the impact of the patterning of oppositely charged residues on the conformational distributions of intrinsically disordered proteins. *Journal of the American Chemical Society* **141**, 4908-4918 (2019).
- 11 Christofi, E. & Barran, P. Ion Mobility Mass Spectrometry (IM-MS) for structural biology: Insights gained by measuring mass, charge, and collision cross section. *Chemical Reviews* **123**, 2902-2949 (2023).

Decision Letter, first revision:

Dear Hongwei,

Thank you for submitting your revised manuscript "Electrospray-Assisted Cryo-EM Sample Preparation to Mitigate Interfacial Effects" (NMETH-A53054A). It has now been seen by the original referees and their comments are below. Thank you also for sending your plan for addressing the concerns that were raised during this round of review. The reviewers find that the paper has improved in revision, and therefore we'll be happy in principle to publish it in Nature Methods, pending the proposed revisions to satisfy the referees' final requests and to comply with our editorial and formatting guidelines.

TRANSPARENT PEER REVIEW

Please note: we allow redactions to authors' rebuttal and reviewer comments in the interest of confidentiality. If you are concerned about the release of confidential data, please let us know specifically what information you would like to have removed. Please note that we cannot incorporate redactions for any other reasons. Reviewer names will be published in the peer review files if the reviewer signed the comments to authors, or if reviewers explicitly agree to release their name. For more information, please refer to our FAQ page.

ORCID

Sincerely,
Arunima

Arunima Singh, Ph.D.

Senior Editor
Nature Methods

Reviewer #1 (Remarks to the Author):

The revision addressed my comments adequately. I have no further comments.

Reviewer #2 (Remarks to the Author):

This manuscript is much improved, and my major questions and reservations have been addressed. The method is surprisingly good and is a promising new approach to cryoEM specimen preparation. In some places of the manuscript the descriptions still need improvement.

For me the key to the method, the principle by which macromolecules are protected from the air-water and substrate-water interfaces, is the creation of multiply-charged cations in the electrospray process. Double layers at the interfaces then repel these cations, keeping them in the middle of the specimen. Unfortunately it was quite late in the manuscript (line 277) where this finally became clear to me; perhaps this mechanism could be explained earlier. It is also very surprising that the highly-charged (highly protonated) macromolecules maintain well their 3D structure, as evidenced by the high-resolution reconstructions. Maybe you could comment on this.

p. 5 line 127. "reduction in the flow rate by 100 nL/min" should be "reduction in the flow rate *to* 100 nL/min"

p. 5, line 163. "Light-proof" is a confusing term, especially as we are talking about electron, not light, penetration. "very thick" would be clearer.

p. 6, line 158. "Nearly comprehensive coverage"; also line 327, "uniform sampling". From Fig. 5d it appears that in each case the angular coverage is more complete than the control, but it is far from "uniform" or "comprehensive" in three of the five cases. Actually, in this figure it is difficult to compare the plots as the upper row (ESI results) seem to be plotted with higher density than the lower row; especially for apo-ferritin and SA the upper panels are much darker.

Fig. 5. Several parts of this figure are confusing.

a) What is the difference between the "positive" and "negative" histograms? Why two histograms for each specimen type? Did you see two separate peaks, one at each interface? Also in part (a), what do "negative" and "positive" refer to? Do positive z-values corresponds to moving closer to an interface, for example?

b) It took me some time to understand that the plots on the right-hand side are all ESI data.

c) I received no additional insight from this part of the figure, it seems just confusing.

p. 7, line 225. What is "Least Squares plane fitting"? This sounds like a very special procedure, but you cite an old, generic review of least squares algorithms. Please explain what it is you did. I take it you fitted a plane to the 3D distribution of particles in the ice layer, and use the fitted z value as the

origin of the z-values of the histograms, right?

p. 7, line 231. What is the "kernel density estimation" (KDE) method? Is this what I just described above? This also comes up on p. 12, line 436.

p. 10, line 358. "diluted...at a concentration of 0.5-4 μM ". I assume that you mean that it was diluted *to* a concentration of 0.5-4 μM , right?

p. 10, line 362. You finally mention the values for the landing distance here, to be 1 to 1.5 cm. I would have been much happier to have known roughly the magnitude of this spacing much earlier, say at line 137.

p, 11, lines 393-395. "...movies were down-sampled....and then motion-corrected using MotionCor2..." This seems to be the wrong sequence of events. I think MotionCor2 does the downsampling *with* the motion correction. That is, MotionCor2 corrects motion using the superresolution data, then downsamples and returns the downsampled result.

Fig. 6a. The arrow near the bottom right, labeled "Spray", pointing to a large droplet picture. What is this supposed to mean? Maybe delete these.

Fig. 6 b. This diagram caused me confusion last time, and I think is likely to mislead readers, as the electrostatics of establishing the protein layer in the specimen is not directly established by his principle. (As I noted last time, the field set up between the tip and the grid is far too small to influence the location of the protein layer.)

Ext. Data Fig. 1. It would be helpful to note which values of the parameters you wound up using. Are they indicated by the dark blue bars, for example?

Ext. Data Fig. 2. Are these results also for apo-ferritin?

Ext. Data Fig. 6d. Why are no percentages given for classes of particles in this part?

-- Fred Sigworth

Reviewer #3 (Remarks to the Author):

Authors have addressed most of my comments. The only concern is the reported resolutions of 70S ribosome, 20S proteasome and apoferritin that were vitrified using standard methods in Extended Data Fig.4. The reported resolutions seem quite low for these well characterized samples. When comparing resolutions between samples vitrified using the ESI method and standard blotting method, have authors controlled for particle number going into the final reconstruction? Using the ESI method, clearly the Euler angle distribution has improved for all specimens. But for the apparent improvement in global resolution using the ESI method to be considered significant, authors would need to compare resolutions for blotting method and ESI method with equal particle numbers that have a reasonably comparable spread of defocus values.

Author Rebuttal, first revision:

Reviewer #2:**Remarks to the Author:**

This manuscript is much improved, and my major questions and reservations have been addressed. The method is surprisingly good and is a promising new approach to cryoEM specimen preparation. In some places of the manuscript, the descriptions still need improvement.

For me the key to the method, the principle by which macromolecules are protected from the air-water and substrate-water interfaces, is the creation of multiply-charged cations in the electrospray process. Double layers at the interfaces then repel these cations, keeping them in the middle of the specimen. Unfortunately it was quite late in the manuscript (line 277) where this finally became clear to me; perhaps this mechanism could be explained earlier.

Response:

We concur with the Reviewer's suggestion regarding the placement of the theoretical description of droplet formation and charge distribution and have revised in the manuscript.

It is also very surprising that the highly-charged (highly protonated) macromolecules maintain well their 3D structure, as evidenced by the high-resolution reconstructions. Maybe you could comment on this.

In ESI of mass spectrometry, analytes transition from the condensed phase to the gas phase and undergo ionization. Positive ion mode allows basic sites in proteins to accept protons. Prior research assumed that the charged water droplet produced in the final Coulomb fission event, containing a neutral protein molecule, would be slightly larger than the protein. Consequently, during the final solvent evaporation, all droplet charges would transfer to the protein. In our demonstration, where proteins are fully immersed in the buffer, they display reduced protonation compared to their gaseous ionic state. Protein protonation induces their repulsion from droplet surfaces, preserving high-resolution features under the high voltage conditions of ESI.

*p. 5 line 127. "reduction in the flow rate by 100 nL/min" should be "reduction in the flow rate *to* 100 nL/min"*

Response:

Corrected.

p. 5, line 163. "Light-proof" is a confusing term, especially as we are talking about electron, not light, penetration. "very thick" would be clearer.

Response:

Corrected.

p. 6, line 158. "Nearly comprehensive coverage"; also line 327, "uniform sampling". From Fig. 5d it appears that in each case the angular coverage is more complete than the control, but it is far from "uniform" or "comprehensive" in three of the five cases.

Response:

We thank the Reviewer for pointing out this incorrect description in the original version. We've rephrased the expression in the manuscript.

Actually, in this figure it is difficult to compare the plots as the upper row (ESI results) seem to be plotted with higher density than the lower row; especially for apo-ferritin and SA the upper panels are much darker.

Response:

The control data (70S ribosome, 20S proteasome, and apoferritin) were derived from previous lab research on rGO supporting films, utilizing a smaller dataset. Both the control and ESI-cryoPrep data employed the entire dataset for 3D reconstruction and angular distribution statistics in previous work. In the revised manuscript, we enhanced visualization by comparing the angular distribution and resolution of the ESI-cryoPrep and control datasets with matched particle counts (Figure R2-1). ESI-cryoPrep data subsets were randomly resampled using subset-generations in the `b_factor_plot.py`, supported by RELION.

Figure R2-1: Angular orientations of ESI-cryoPrep and control datasets with matched particle counts.

Fig. 5. Several parts of this figure are confusing.

a) What is the difference between the "positive" and "negative" histograms? Why two histograms for each specimen type? Did you see two separate peaks, one at each interface? Also in part (a), what do "negative" and "positive" refer to? Do positive z-values correspond to moving closer to an interface, for example?

b) It took me some time to understand that the plots on the right-hand side are all ESI data.

c) I received no additional insight from this part of the figure, it seems just confusing.

Response:

Figure 5 provides a statistical supplement to the investigation of particle spatial distribution in cryo-ET. To enhance statistical robustness, we employed a larger dataset, conducting a comprehensive distribution analysis within the complete single-particle analysis dataset by evaluating relative Z-heights derived from defocus variations.

In Figure 5a, the green color panel displayed a bimodal distributions of the z-height of apo-ferritin particles on supporting films suggest simultaneous protein adsorption at both air-water and graphene-water interfaces. This distribution was not observed in ESI-cryoPrep data. The terms "positive" and "negative" correspond to the signs of skewness values. In probability theory and statistics, skewness measures the asymmetry of the probability distribution of a real-valued random variable about its mean. For a unimodal distribution, a negative skew commonly indicates that the tail is on the left side, while a positive skew indicates that the tail is on the right. If the distribution is symmetric and unimodal, the mean equals the median, resulting in zero skewness. Positive skewness suggests that most particles are located on the side with a smaller absolute defocus value, farther from the objective lens, and conversely.

p. 7, line 225. What is "Least Squares plane fitting"? This sounds like a very special procedure,

but you cite an old, generic review of least squares algorithms. Please explain what it is you did. I take it you fitted a plane to the 3D distribution of particles in the ice layer, and use the fitted z value as the origin of the z-values of the histograms, right?

Response:

Yes, you are absolutely right.

Due to uneven flatness in the ice surface and EM grid, a certain level of inclination is present in particle distribution. At the micrograph scale, particles tend to align along a flat surface, exhibiting a slight angular deviation from the horizontal plane. Calculating particle distances from the fitting plane corrects inaccuracies stemming from small angular tilts, providing a more precise evaluation of distribution patterns. The least-squares plane-fitting method enables the observation of various distribution patterns, including bilayer particle distributions, which were undistinguishable in coarse analyses. Validation with tilt data confirms that the tilt of the fitted plane aligns with the tilt angle.

p. 7, line 231. What is the "kernel density estimation" (KDE) method? Is this what I just described above? This also comes up on p. 12, line 436.

Response:

Kernel Density Estimation (KDE) is a non-parametric statistical technique used to estimate the probability density function (PDF) of a random variable. It provides a smooth, continuous representation of the underlying distribution of data points without assuming a specific mathematical form for the distribution.

In KDE, a kernel function is placed at each data point, and these kernels are summed or averaged to create a smooth estimate of the probability density. The width of the kernel, known as the bandwidth, determines the smoothness of the estimated density.

Mathematically, the kernel density estimate $\hat{f}(x)$ at a point x is given by:

$$\hat{f}(x) = \frac{1}{nh} \sum_{i=1}^n K\left(\frac{x - x_i}{h}\right)$$

Here, n is the number of data points, h is the bandwidth, x_i are the data points, and K is the kernel function.

Demonstration:

Consider a dataset of observations. To estimate its probability density using KDE, you would place a kernel at each data point, sum them up, and normalize to obtain a smooth density curve. The KDE function can be called from within Python.

*p. 10, line 358. "diluted...at a concentration of 0.5-4 μM ". I assume that you mean that it was diluted *to* a concentration of 0.5-4 μM , right?*

Response:

Corrected.

p. 10, line 362. You finally mention the values for the landing distance here, to be 1 to 1.5 cm. I would have been much happier to have known roughly the magnitude of this spacing much earlier, say at line 137.

Response:

We appreciate the Reviewer for the suggestion and have revised in the manuscript.

*p, 11, lines 393-395. "...movies were down-sampled....and then motion-corrected using MotionCor2..." This seems to be the wrong sequence of events. I think MotionCor2 does the downsampling *with* the motion correction. That is, MotionCor2 corrects motion using the superresolution data, then downsamples and returns the downsampled result.*

Response:

We thank the Reviewer for pointing out this error and have corrected it in the revised version of the manuscript.

Fig. 6a. The arrow near the bottom right, labeled "Spray", pointing to a large droplet picture. What is this supposed to mean? Maybe delete these.

Response:

We appreciate the Reviewer for the suggestion and have corrected it in the revised version of the manuscript.

Fig. 6 b. This diagram caused me confusion last time, and I think is likely to mislead readers, as the electrostatics of establishing the protein layer in the specimen is not directly established by his principle. (As I noted last time, the field set up between the tip and the grid is far too small to influence the location of the protein layer.)

Response:

We have removed the Fig. 6b to eliminate ambiguity in the revised manuscript.

Ext. Data Fig. 1. It would be helpful to note which values of the parameters you wound up using. Are they indicated by the dark blue bars, for example?

Response:

We appreciate the guidance provided by the Reviewer and have addressed this in the revised Ext. Data Fig. 1.

Ext. Data Fig. 2. Are these results also for apo-ferritin?

Response:

Yes, mass spectrometry data is obtained using apo-ferritin. The favorable conditions identified in EM align with those observed in native MS analysis.

Ext. Data Fig. 6d. Why are no percentages given for classes of particles in this part?

Response:

We have revised Ext. Data Fig. 6d to include percentages of classes in 3D classification.

-- Fred Sigworth

Reviewer #3:

Authors have addressed most of my comments. The only concern is the reported resolutions of 70S ribosome, 20S proteasome, and apoferritin that were vitrified using standard methods in Extended Data Fig. 4. The reported resolutions seem quite low for these well-characterized samples. When comparing resolutions between samples vitrified using the ESI method and standard blotting method, have authors controlled for particle number going into the final reconstruction? Using the ESI method, clearly the Euler angle distribution has improved for all specimens. But for the apparent improvement in global resolution using the ESI method to be considered significant, authors would need to compare resolutions for blotting method and ESI method with equal particle numbers that have a reasonably comparable spread of defocus values.

Response:

The control data (70S ribosome, 20S proteasome, and apoferritin) were derived from prior laboratory investigations using rGO supporting films with a smaller dataset. Both the control and ESI-cryoPrep datasets employed the entire dataset for 3D reconstruction and angular distribution statistics.

When conducting 3D reconstruction with experimental and control data at similar particle levels, the ESI-cryoPrep method exhibits significantly higher resolution of the 3D density maps compared to the control, even when analyzing a small portion of the original dataset. Angular orientations were characterized in response to Reviewer 2. The 3D-FSC analysis of the four ESI-cryoPrep specimens and control data with matched particle counts is presented below (Figure R3-2).

Figure R3-1: Gold-standard and model-to-map FSC of ESI-cryoPrep and control datasets with matched particle counts.

Figure R3-2: 3DFSC analysis of ESI-cryoPrep and control datasets with complete and matched particle counts.

We then took an additional larger dataset of 20S proteasome prepared using rGO supporting films, collected on the same Titan Krios alongside the ESI-cryoPrep dataset, and captured at the same magnification for comparative analysis (Figure R3-3). The entire 20S proteasome control dataset, consisting of 56,723 particles, was reconstructed to a final resolution of 2.78 Å, while the ESI-cryoPrep dataset, with a comparable particle count, was reconstructed to 2.25 Å.

Figure R3-3: 3DFSC analysis of ESI-cryoPrep and larger control datasets with matched particle counts.

P.S. All 3DFSC data were computed using the sharpened density map and mask.

Final Decision Letter:

Dear Hongwei,

I am pleased to inform you that your Article, "Electrospray-Assisted Cryo-EM Sample Preparation to Mitigate Interfacial Effects", has now been accepted for publication in Nature Methods. The received and accepted dates will be July 3, 2023 and March 17, 2024. This note is intended to let you know what to expect from us over the next month or so, and to let you know where to address any further questions.

Over the next few weeks, your paper will be copyedited to ensure that it conforms to Nature Methods style. Once your paper is typeset, you will receive an email with a link to choose the appropriate publishing options for your paper and our Author Services team will be in touch regarding any additional information that may be required. It is extremely important that you let us know now whether you will be difficult to contact over the next month. If this is the case, we ask that you send us the contact information (email, phone and fax) of someone who will be able to check the proofs and deal with any last-minute problems.

Please note that *Nature Methods* is a Transformative Journal (TJ). Authors may publish their research with us through the traditional subscription access route or make their paper immediately open access through payment of an article-processing charge (APC). Authors will not be required to make a final decision about access to their article until it has been accepted. Find out more about Transformative Journals

You may wish to make your media relations office aware of your accepted publication, in case they consider it appropriate to organize some internal or external publicity. Once your paper has been scheduled you will receive an email confirming the publication details. This is normally 3-4 working days in advance of publication. If you need additional notice of the date and time of publication,

please let the production team know when you receive the proof of your article to ensure there is sufficient time to coordinate. Further information on our embargo policies can be found here: <https://www.nature.com/authors/policies/embargo.html>

If you are active on Twitter/X, please e-mail me your and your coauthors' handles so that we may tag you when the paper is published.

Best regards,
Arunima

Arunima Singh, Ph.D.
Senior Editor
Nature Methods